# VARestorer: One-Step VAR Distillation for Real-World Image Super-Resolution

**Yixuan Zhu, Shilin Ma, Haolin Wang, Ao Li,**
**Yanzhe Jing, Yansong Tang✉, Lei Chen, Jiwen Lu, Jie Zhou**
Tsinghua University

## Abstract

Recent advancements in visual autoregressive models (VAR) have demonstrated their effectiveness in image generation, highlighting their potential for real-world image super-resolution (Real-ISR). However, adapting VAR for ISR presents critical challenges. The next-scale prediction mechanism, constrained by causal attention, fails to fully exploit global low-quality (LQ) context, resulting in blurry and inconsistent high-quality (HQ) outputs. Additionally, error accumulation in the iterative prediction severely degrades coherence in ISR task. To address these issues, we propose VARestorer, a simple yet effective distillation framework that transforms a pre-trained text-to-image VAR model into a one-step ISR model. By leveraging distribution matching, our method eliminates the need for iterative refinement, significantly reducing error propagation and inference time. Furthermore, we introduce pyramid image conditioning with cross-scale attention, which enables bidirectional scale-wise interactions and fully utilizes the input image information while adapting to the autoregressive mechanism. This prevents later LQ tokens from being overlooked in the transformer. By fine-tuning only 1.2% of the model parameters through parameter-efficient adapters, our method maintains the expressive power of the original VAR model while significantly enhancing efficiency. Extensive experiments show that VARestorer achieves state-of-the-art performance with 72.32 MUSIQ and 0.7669 CLIPIQA on DIV2K dataset, while accelerating inference by 10 times compared to conventional VAR inference. Our code is available at https://github.com/EternalEvan/VARestorer.

## 1 Introduction

Real-world image super-resolution (Real-ISR) aims to enhance visual quality, as images captured in the wild suffer from noise, blur, downsampling, and compression due to device limitations and complex environments. In recent years, great development has been achieved in the image enhancement field using deep learning methods (Dong et al., 2014; Zhang et al., 2017; Liang et al., 2021; Chen et al., 2021; Lin et al., 2023; Zamir et al., 2022; Chen et al., 2023). While these methods yield commendable outcomes under specific, well-defined degradations, they often fall short when faced with the complex conditions of real-world scenarios. Thus, our basic goal is to construct an effective and robust image enhancement framework capable of addressing various degradation conditions. This framework aims to deliver high-quality, visually appealing, and structurally consistent results within a limited computational budget, making it more practical for diverse real-world use cases.

Real-ISR is inherently ill-posed due to unknown degradation processes, allowing for various possible high-quality (HQ) results from low-quality (LQ) inputs. To address this problem, researchers have explored a wide range of deep learning methods, which can be generally categorized into three classes: predictive (Huang et al., 2020; Gu et al., 2019; Zhang et al., 2018a), GAN-based (Wang et al., 2021a; Yuan et al., 2018; Fritsche et al., 2019; Zhang et al., 2021b; Wang et al., 2021c) and diffusion-based (Kawar et al., 2022; Wang et al., 2022; Fei et al., 2023; Lin et al., 2023; Yue et al., 2023). Predictive methods estimate blur kernels via convolutional networks but struggle in real-world conditions. To better handle real-world challenges, some approaches leverage Generative Adversarial Networks (GANs) (Goodfellow et al., 2014) to jointly learn the data distribution of the images and various degradation types. GAN-based methods significantly enhance image quality but require careful tuning of sensitive hyper-parameters during training. In recent years, diffusion

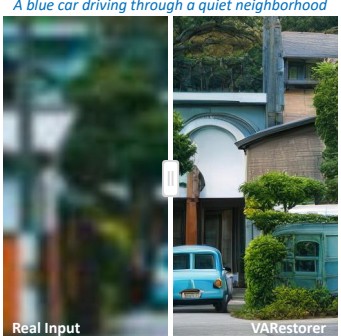 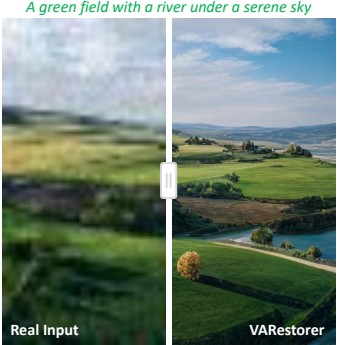 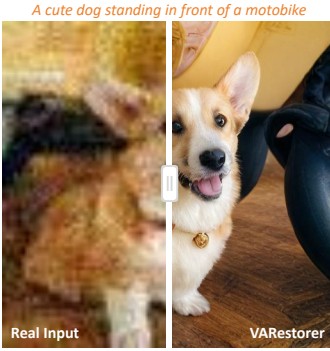

Figure 1: Our VARestorer showcases remarkable image restoration capabilities across complex degradations. With a highly effective VAR distillation framework, VARestorer adeptly harnesses the rich knowledge within the pre-trained VAR model for real-world image super-resolution in a single step. Full input-output comparisons are provided in the supplementary material.

models (DMs) (Ho et al., 2020; Rombach et al., 2022) have shown impressive visual generation capacity for image synthesis tasks. Some methods (Yu et al., 2024a; Wu et al., 2024a; Lin et al., 2023; Yue et al., 2023) adopt the pre-trained diffusion models and restore the images during the denoising sampling. Most recently, visual autoregressive models (VAR) (Tian et al., 2025) arise as a powerful image synthesis framework with the next-scale prediction mechanism, which is inherently potential for super-resolution. However, the error accumulation and the sequential prediction nature of VAR model harm its performance to generate a consistent and high-quality result.

To address the aforementioned challenges, we propose a novel VAR distillation framework that distills a pre-trained VAR model into an efficient one-step model for real-world image super-resolution. To maintain the image quality of VAR models, we leverage the token-level distribution matching to align the generation quality of the one-step student model and the pre-trained VAR. Our approach eliminates error accumulation during iterative sampling and significantly accelerates inference, while fully utilizing the learned generative priors of VAR, as depicted in Figure 2. Moreover, we devise the cross-scale pyramid conditioning mechanism to fully leverage the information of low and high scales. This also preserves the original architecture and capabilities of VAR, reducing the difficulty of VAR distillation. After the VAR distillation, our framework establishes a direct one-step mapping from low-quality inputs to high-quality results. This mapping is carefully optimized to ensure that the output images maintain the fidelity and realism of the teacher model while distinctly diverging from undesirable visible artifacts.

We comprehensively evaluate our framework on both real-world and synthetic datasets. Experimental results underscore the proficiency of our VARestorer across these datasets. For the synthesis dataset, VARestorer achieves 72.32 MUSIQ and 0.7669 CLIPIQA on the synthetic DIV2K-Val. Additionally, it sets new benchmarks with an NIQE of 4.41, highlighting its ability to restore high-fidelity textures. For real-world datasets, we attain 0.5638 and 0.5655 MANIQA on the DrealSR and RealSR datasets, demonstrating both high restoration quality and efficiency in a single step. VARestorer consistently delivers visually appealing, perceptually convincing, and semantically plausible enhancements, underscoring the proficiency of our framework, as illustrated in Figure 1.

## 2 RELATED WORKS

**Real-world image super-resolution.** Real-world image super-resolution involves tasks like denoising, deblurring, and super-resolution, *etc*. Conventional works (Dong et al., 2014; Huang et al., 2020; Zhang et al., 2018a; Dong et al., 2015) utilize predictive models to estimate blur kernels and restore HQ images. With the rise of vision transformers (Dosovitskiy et al., 2020; Liu et al., 2021), some methods (Chen et al., 2021) incorporate the attention mechanism into basic architectures, yielding high-quality results. However, these models struggle with real-world degradations. The advent of generative models has introduced two main approaches in real-ISR, achieving significant success in complex blind image restoration tasks. One approach is GAN-based methods (Wang et al., 2021a; Yuan et al., 2018; Fritsche et al., 2019; Zhang et al., 2021b; Wang et al., 2021c;b; Yang et al., 2021; Zhou et al., 2022), which process images in the latent space to handle tasks like

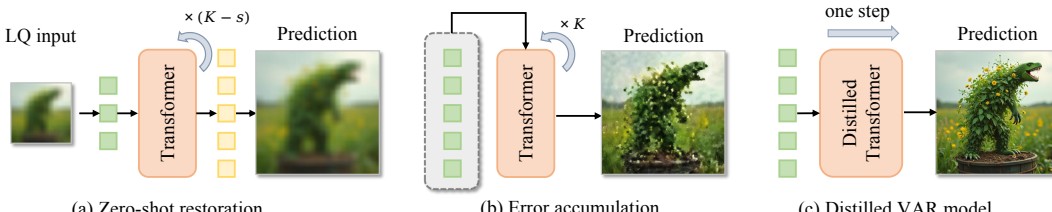

Figure 2: Comparison of VAR-based ISR approaches. (a) Zero-shot upsampling uses LQ tokens at scale $s$, but severe degradations limit effectiveness. (b) Image-conditioned VAR improves clarity but suffers from error accumulation and artifacts. (c) VARestorer distills VAR into a one-step model, minimizing errors while preserving generative capability without extra conditioning.

real-ISR. However, GAN-based methods require meticulous hyper-parameter tuning and they are often tailored to specific tasks, limiting their versatility. The other approach involves diffusion models (Ho et al., 2020; Rombach et al., 2022), known for their impressive image generation capabilities. Methods like (Wang et al., 2023b; Yue & Loy, 2022; Yue et al., 2023; Kawar et al., 2022; Fei et al., 2023; Lin et al., 2023; Yu et al., 2024a) design specific denoising structures to transfer the image generation framework to image restoration tasks. However, the sampling of diffusion models is time-consuming. To address this, some methods like (Xie et al., 2024; Wang et al., 2024; Zhu et al., 2024; Wu et al., 2024a; Zhu et al., 2025) employ diffusion distillation frameworks to process images in fewer steps. Nevertheless, distillation results of diffusion models often suffer from oversmoothing and reduced diversity, especially when facing complex degeneration.

**Visual autoregressive models.** Building on the success of LLMs (Touvron et al., 2023a;b; Brown et al., 2020; Radford et al., 2019), autoregressive models adopt discrete quantizers such as VQ-VAE (Van Den Oord et al., 2017) to encode image patches into tokenized representations, enabling image generation through next-token prediction (Van Den Oord et al., 2017; Esser et al., 2021; Razavi et al., 2019). However, the sequential nature of flattened token prediction can disrupt spatial coherence and structure consistency. Recently, visual autoregressive models (VAR) (Tian et al., 2025) shift from the next-token prediction to the next-scale prediction, greatly improving image generation quality while ensuring superior scalability. VAR-based methods (Han et al., 2024; Li et al., 2024b; Ma et al., 2024) have expanded to other conditional generation tasks (e.g., C2I, T2I), achieving results comparable to diffusion models. Despite this progress, VAR remains underexplored for super-resolution due to limited cross-scale dependency modeling, error accumulation, and less effective conditioning (Li et al., 2024a; Yao et al., 2024). To overcome these issues, we propose an effective VAR distillation framework to minimize error accumulation. We also design the cross-scale pyramid conditioning which not only preserves the generative capability of VAR but also enables bidirectional attention across scales.

## 3 METHOD

In this section, we present VARestorer, a one-step VAR-based framework that achieves real-world image super-resolution with distribution matching distillation of VAR models. Our key idea is to investigate the inherent knowledge in a pre-trained text-to-image VAR model and minimize the error accumulation with single-step inference for real-ISR. We will start by reviewing the background of visual autoregressive models, and then describe our designs of VARestorer, including one-step VAR distillation via distribution matching and the cross-scale pyramid conditioning for fully leveraging the LQ input information. The overall framework of our VARestorer is illustrated in Figure 3.

### 3.1 PRELIMINARY: VISUAL AUTOREGRESSIVE MODELS.

Autoregressive models formulate data generation as a next-token prediction process, where a data sample $\boldsymbol{x} = (\boldsymbol{x}_1, \boldsymbol{x}_2, \ldots, \boldsymbol{x}_T)$ is modeled as a product of conditional probabilities:

$$p(\boldsymbol{x}) = \prod_{t=1}^{T} p(\boldsymbol{x}_t \mid \boldsymbol{x}_1, \boldsymbol{x}_2, \ldots, \boldsymbol{x}_{t-1}), \tag{1}$$

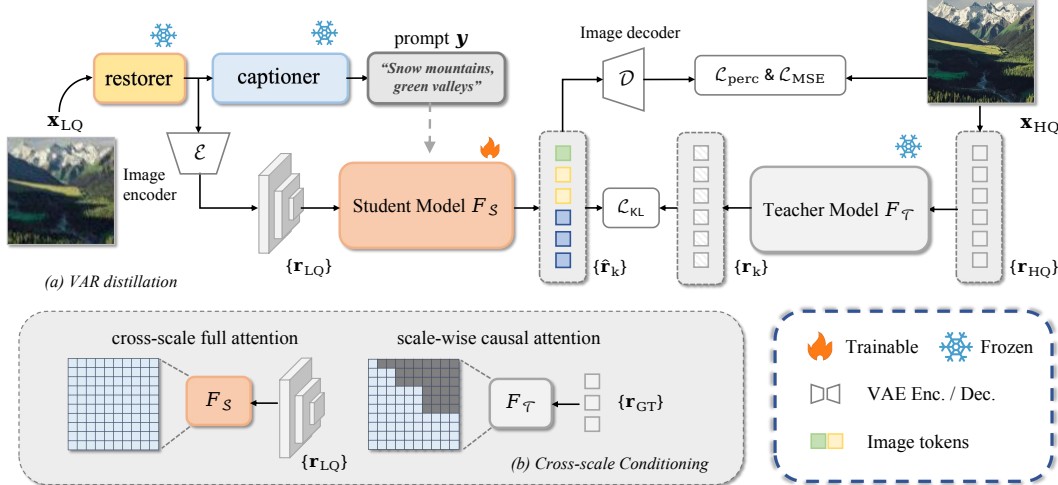

Figure 3: **The overall framework of *VARestorer*.** (a) VARestorer utilizes VAR distillation framework for real-ISR. During training, we employ the pre-trained text-to-image VAR model as the teacher to predict the high-quality tokens and calculate the token-level KL divergence for distribution alignment. (b) To fully exploit the LQ input, we introduce cross-scale pyramid conditioning, which allows the student model to use full attention to learn relationships among features produced by the multi-scale VAE. During inference, VARestorer produces high-quality results in a single step.

where $\boldsymbol{x}_t$ is the token at step $t$. AR models have been widely used in sequence modeling tasks, such as natural language processing and generative modeling. However, their sequential nature makes them computationally expensive and prone to error accumulation over long sequences.

Visual autoregressive (VAR) models extend the AR framework for image generation by introducing next-scale prediction mechanism, where an image $\boldsymbol{x} \in \mathbb{R}^{H \times W \times C}$ is represented as a sequence of discrete token maps at different scales $\boldsymbol{x} = (\boldsymbol{r}_1, \boldsymbol{r}_2, \ldots, \boldsymbol{r}_K)$. The generation process follows a sequential dependency across scales:

$$p(\boldsymbol{x}) = \prod_{k=1}^{K} p(\boldsymbol{r}_k \mid \boldsymbol{r}_1, \boldsymbol{r}_2, \ldots, \boldsymbol{r}_{k-1}), \tag{2}$$

where $\boldsymbol{r}_k \in [V]^{h_k \times w_k}$ is the token map at scale $k$, with dimensions $h_k$ and $w_k$, conditioned on previous scales $(\boldsymbol{r}_1, \boldsymbol{r}_2, \ldots, \boldsymbol{r}_{k-1})$. Each token in $\boldsymbol{r}_k$ is an index from the VQVAE codebook $V$, which is learned via multi-scale quantization and shared across scales.

## 3.2 IMAGE RESTORATION VIA VAR DISTILLATION

The pre-trained VAR model encodes rich information about real-world data distributions and excels at detailed image synthesis. Its next-scale prediction mechanism aligns naturally with super-resolution, making it a promising candidate for image restoration. Our goal is to leverage the generative priors of a pre-trained diffusion model for image restoration while significantly reducing the computational overhead of VAR-based upsampling. To achieve this, we propose an efficient distillation framework with tailored conditioning to transform the autoregressive transformer $\mathcal{F}$ from a text-to-image pre-trained VAR model into a one-step image enhancer. This approach enables direct upsampling, predicting all high-quality image tokens in a single pass.

However, we observe that while the VAR model learns to predict next-scale tokens, it cannot directly upscale real-world images like a dedicated super-resolution model. As shown in Figure 2, we input low-resolution (LR) tokens at scale level $s$ into the pre-trained VAR model and use its autoregressive predictions to complete high-resolution (HR) components. However, this zero-shot approach produces noticeable artifacts and suboptimal results. We hypothesize that this limitation arises because LR tokens fail to provide sufficiently detailed conditioning for the autoregressive transformer to predict finer scales accurately. Additionally, LR tokens from real-world images may not align well with the learned token maps due to other degradations in low-quality (LQ) inputs like noise and

blurring. Another key limitation of VAR models is error accumulation during iterative inference. In text and image generation tasks, AR models can produce suboptimal tokens at intermediate steps as long as the final output remains plausible. However, in image restoration, the model must generate a deterministic output that precisely aligns with the groundtruth. Consequently, iterative sampling in VAR can lead to severe mismatches and artifacts due to error propagation. To address this, we aim to minimize the upsampling process to a single step, eliminating room for error accumulation. This motivates our exploration of VAR distillation, which compresses the rich generative knowledge of VAR into a lightweight one-step model tailored for real-world image restoration.

Since our goal is to distill a multi-step generative model to a one-step model, we follow the previous diffusion distillation method like (Yin et al., 2023; Nguyen & Tran, 2024) to incorporate the distribution matching framework to transfer the generative knowledge from teacher model $\mathcal{T}$—a pre-trained VAR model—to the student one-step model $\mathcal{S}$. This process can be formulated as an optimization problem for the KL divergence between real image distribution $p_t$ and generated distribution $q_t$, $D_{\mathrm{KL}}(p_t\|q_t)$. In diffusion models, the density of the data distribution can be estimated by the denoising model (Song et al., 2020), allowing the KL gradient to be estimated via two denoising models. However, due to the fundamental differences in image and distribution formulations between VAR and diffusion models, a direct application of this approach is impractical. Instead, we explore an alternative pathway to solve this optimization, starting with the KL divergence formulation in the VAR model based on predicted token distributions across scales:

$$\mathcal{L}_{\mathrm{KL}} = \sum_k D_{\mathrm{KL}}(p_{\mathcal{T}}(\boldsymbol{r}_k \mid \boldsymbol{r}_{\mathrm{HQ},<k}) \,\|\, p_{\mathcal{S}}(\hat{\boldsymbol{r}}_k \mid \boldsymbol{r}_{\mathrm{LQ}})), \tag{3}$$

where $\boldsymbol{r}_{\mathrm{HQ},<k}$ is the GT token maps before scale $k$. $\hat{\boldsymbol{r}}_k$ is the student's predicted tokens and $\boldsymbol{r}_{\mathrm{LQ}}$ is the token maps of the input LQ image. The teacher model generates high-resolution tokens $\boldsymbol{r}_k$ sequentially from low-resolution tokens $\boldsymbol{r}_{\mathrm{HQ},<k}$, while our one-step student model directly predicts all tokens at once given the LQ image input:

$$p_{\mathcal{T}}(\boldsymbol{r}_k \mid \boldsymbol{r}_{\mathrm{HQ},<k}) = \mathcal{F}_{\mathcal{T}}(\boldsymbol{r}_{\mathrm{HQ},<k}), \; p_{\mathcal{S}}(\hat{\boldsymbol{r}} \mid \boldsymbol{r}_{\mathrm{LQ}}) = \mathcal{F}_{\mathcal{S}}(\boldsymbol{r}_{\mathrm{LQ}}). \tag{4}$$

With token-level distribution matching, we force the student's token predictions at different scales aligned with the teacher model. This ensures that the student directly learns to generate high-resolution tokens in a single pass, mimicking the teacher's step-by-step generation process. Unlike pixel-wise losses, KL loss encourages the student model to learn diverse, high-quality predictions rather than just averaging outputs. Besides distribution matching, we also adopt the widely used perception and MSE loss to improve the consistency between the predicted image $\boldsymbol{x}_{\mathcal{S}} = (\hat{\boldsymbol{r}}_1, \hat{\boldsymbol{r}}_2, \ldots, \hat{\boldsymbol{r}}_K)$ and the groundtruth $\boldsymbol{x}_{\mathrm{GT}}$. The final loss function combines token-level KL alignment with additional consistency losses:

$$\mathcal{L} = \lambda_{\mathrm{KL}}\mathcal{L}_{\mathrm{KL}} + \lambda\mathrm{perc}\mathcal{L}\mathrm{perc} + \lambda_{\mathrm{MSE}} \|\boldsymbol{x}_{\mathcal{S}} - \boldsymbol{x}_{\mathrm{GT}}\|_2^2. \tag{5}$$

By optimizing these terms, the student model learns to enhance image quality, fidelity, and consistency in an end-to-end manner. Once trained, it enables one-step inference given LQ input, significantly accelerating the traditional VAR prediction process.

### 3.3 CROSS-SCALE PYRAMID CONDITIONING

We then explore how to effectively incorporate the low-quality (LQ) image into the student model. In diffusion-based models, various image conditioning strategies have been developed to embed image information into the denoising process, thereby guiding the denoising trajectory. A straightforward approach is to concatenate the control image with the noisy image. Another effective method is ControlNet (Zhang et al., 2023), which injects conditioning features into the intermediate layers of the model. These strategies can be adapted similarly to the VAR model. However, a fundamental challenge arises due to VAR's hierarchical token prediction: *How many control tokens should be used at each scale?* A naive solution is to maintain the number of control tokens equal to $\boldsymbol{r}_k$ at each scale $k$ to match the pre-trained model and merge these control tokens with $\boldsymbol{r}_k$. However, this limits the ability to fully exploit the input image's information, particularly at lower scales, where the guidance signal becomes ineffective. While ControlNet offers a powerful way to inject detailed image features, prior works like (Yao et al., 2024) and (Li et al., 2024a) have shown that directly applying ControlNet to VAR can disrupt the autoregressive generation rather than enhancing it, requiring extensive modifications and retraining.

To address this, we propose cross-scale pyramid conditioning, inspired by VAR's zero-shot upsampling. Instead of modifying the model's architecture extensively, we finetune the VAE encoder of the pre-trained VAR model to generate multi-scale token maps, forming a pyramid representation of the input image. Each level of this pyramid captures different levels of detail, ensuring both high-level semantics and fine-grained structures are effectively utilized. To fully exploit these tokens, we modify the causal attention mask in VAR to allow full attention across scales. This modification enables direct interaction between all resolution levels, ensuring that the model retains its generative prior while making full use of the conditioning information. By employing this strategy, our method preserves the original architecture and capabilities of the pre-trained VAR model, while significantly improving its ability to handle real-world degradations and achieve high-quality restoration.

## 3.4 IMPLEMENTATION

To leverage VAR's generative power, we initialize both student $\mathcal{S}$ and teacher $\mathcal{T}$ models with a pre-trained text-to-image VAR (Han et al., 2024). In order to construct a pyramid representation of the input image, we fine-tune the VAE encoder of the student model, enabling the extraction of high-quality multi-scale token maps. We use BLIP (Li et al., 2022) to generate textual prompts, enabling the model to exploit pre-trained vision-language knowledge. With the textual prompt as an additional hint, the model can better exploit the rich pre-trained knowledge of image understanding and vision-language relationships acquired during the generation task. To tune the student model, we unfreeze the cross-attention layers of $\mathcal{S}$ by LoRA (Hu et al., 2022), which consists of merely 1.2% trainable parameters of the transformer, and freeze other parameters. For the restorer, we employ the lightweight module in (Liang et al., 2021) following (Lin et al., 2023) to perform a coarse restoration. To approximate the degradation conditions in BFR and BSR, we produce the synthetic data from HQ image by $x_{\text{LQ}} = [(k * x_{\text{HQ}}) \downarrow_r + n]_{\text{JPEG}}$, which consists of blur, noise, resize and JPEG compression. Further details are provided in Section A.

## 4 EXPERIMENTS

We conduct experiments across various datasets to verify the effectiveness of our method. We will start by describing the experimental settings and then present our main quantitative and qualitative results. We will also provide detailed ablation studies and in-depth analyses of our approach to highlight each component's contribution and validate our overall design choices.

## 4.1 EXPERIMENT SETUPS

**Datasets.** For a fair comparison, we follow previous work (Wu et al., 2024a) to establish our training and evaluation datasets. For simplicity, we train our model using the LSDIR (Li et al., 2023) dataset, which consists of about 85,000 high-quality images. We evaluate our model and compare it with competing methods using the test set provided by StableSR (Wang et al., 2023c), including both synthetic and real-world data. The synthetic data includes 3000 images of size $512 \times 512$, whose GTs are randomly cropped from DIV2K-Val (Agustsson & Timofte, 2017) and degraded using the Real-ESRGAN pipeline (Wang et al., 2021c). The real-world data include LQ-HQ pairs from RealSR (Cai et al., 2019) and DRealSR (Wei et al., 2020) to validate performance under genuine degradations.

**Training details.** We unfreeze the cross-attention layers and self-attention layers in the student model $\mathcal{S}$ in our framework during training, which allows the student to better capture both global dependencies and fine-grained interactions across modalities. With a batch size of 32 and a learning rate of 1e-6 using AdamW optimizer with a weight decay of 1e-2, we initialize the student and teacher models by replicating the autoregressive transformer blocks in (Han et al., 2024). For LQ-HQ pair synthesis, we employ the high-order degradation model in (Wang et al., 2021c), training for 10K steps with 8 Nvidia L20 GPUs, respectively. We set the KL term weight $\lambda_{\text{KL}}$ to 0.1, the perception term weight $\lambda_{\text{perc}}$ to 0.25 and the MSE term weight $\lambda_{\text{MSE}}$ to 0.5 for balance and optimal performance. To unfreeze the student model, we utilize LoRA to inject trainable parameters into the cross-attention modules and self-attention modules, with the rank set to 32, thereby enabling efficient adaptation without introducing excessive computational cost.

**Metrics.** To evaluate our VARestorer's performance on image restoration, we calculate three traditional metrics, including PSNR, SSIM and LPIPS (Zhang et al., 2018b). However, these metrics

Table 1: **Quantitative comparison on both synthetic and real-world benchmarks.** The best and second-best values for each metric are highlighted in **red** and **blue**, respectively. The number after each method denotes the inference steps. Our framework achieves high-quality results and outperforms existing methods in various image quality metrics.

| Datasets | Methods | PSNR↑ | SSIM↑ | LPIPS↓ | MANIQA↑ | MUSIQ↑ | NIQE↓ | CLIPIQA↑ | LIQE↑ | QALIGN↑ | FID↓ |
|---|---|---|---|---|---|---|---|---|---|---|---|
| DIV2K-Val | DiffBIR-50 | 21.48 | 0.5050 | 0.3670 | **0.5664** | 69.87 | 5.003 | 0.7303 | **4.346** | **4.070** | 32.75 |
| | SeeSR-50 | 21.97 | 0.5673 | 0.3193 | 0.5036 | 68.67 | 4.808 | 0.6936 | 4.274 | 4.035 | **25.90** |
| | PASD-20 | 22.31 | 0.5675 | 0.3296 | 0.4371 | 67.78 | **4.581** | 0.6459 | 3.947 | 3.895 | 35.47 |
| | ResShift-15 | **22.66** | **0.5888** | **0.3077** | 0.3693 | 58.90 | 6.916 | 0.5715 | 3.082 | 3.309 | 30.81 |
| | OSEDiff-1 | 22.06 | **0.5735** | **0.2942** | 0.4410 | 67.96 | 4.711 | 0.6680 | 4.117 | 3.926 | **26.34** |
| | SinSR-1 | **22.52** | 0.5680 | 0.3240 | 0.4216 | 62.77 | 6.005 | 0.6483 | 3.493 | 3.553 | 35.45 |
| | VARSR-10 | 22.41 | 0.5724 | 0.3177 | 0.5173 | **71.48** | 5.977 | **0.7330** | 4.282 | 3.853 | 33.86 |
| | VARestorer-1 | 21.08 | 0.5355 | 0.3131 | **0.5590** | **72.32** | **4.410** | **0.7669** | **4.664** | **4.363** | 31.11 |
| DrealSR | DiffBIR-50 | 24.05 | 0.5831 | 0.4669 | **0.5543** | 66.14 | 6.329 | 0.7072 | 4.101 | 3.734 | 180.4 |
| | SeeSR-50 | 25.82 | 0.7405 | 0.3174 | 0.5128 | 65.09 | 6.407 | 0.6905 | 4.126 | **3.754** | **147.3** |
| | PASD-20 | **26.14** | **0.7466** | **0.3081** | 0.4404 | 62.34 | **6.126** | 0.6293 | 3.603 | 3.572 | 164.1 |
| | ResShift-15 | 24.48 | 0.6803 | 0.4169 | 0.3232 | 50.77 | 8.941 | 0.5371 | 2.629 | 2.877 | 159.7 |
| | OSEDiff-1 | 25.85 | **0.7548** | **0.2966** | 0.4657 | 64.69 | 6.464 | 0.6962 | 3.939 | 3.746 | **135.4** |
| | SinSR-1 | 25.83 | 0.7157 | 0.3655 | 0.3901 | 55.64 | 6.953 | 0.6447 | 3.131 | 3.135 | 172.7 |
| | VARSR-10 | **26.05** | 0.7353 | 0.3536 | 0.5361 | **68.14** | 6.971 | 0.7215 | **4.137** | 3.480 | 156.5 |
| | VARestorer-1 | 24.31 | 0.6894 | 0.3584 | **0.5638** | 69.49 | **5.494** | **0.7810** | **4.582** | **4.188** | 149.7 |
| RealSR | DiffBIR-50 | 23.33 | 0.6180 | 0.3650 | **0.5583** | 69.28 | 5.839 | **0.7054** | 4.101 | 3.760 | 130.8 |
| | SeeSR-50 | 23.60 | 0.6947 | 0.3007 | 0.5437 | 69.82 | 5.396 | 0.6696 | 4.136 | 3.789 | 125.4 |
| | PASD-20 | **24.83** | **0.7247** | **0.2709** | 0.4423 | 66.93 | **5.349** | 0.5815 | 3.575 | 3.705 | 131.9 |
| | ResShift-15 | 23.67 | 0.6931 | 0.3451 | 0.3538 | 56.90 | 8.331 | 0.5350 | 2.891 | 3.111 | 129.5 |
| | OSEDiff-1 | 23.59 | 0.7074 | **0.2920** | 0.4716 | 69.08 | 5.652 | 0.6685 | 4.070 | **3.801** | **123.5** |
| | SinSR-1 | **24.50** | **0.7076** | 0.3219 | 0.4045 | 61.07 | 6.319 | 0.6178 | 3.200 | 3.299 | 140.8 |
| | VARSR-10 | 24.12 | 0.7001 | 0.3216 | 0.5465 | **71.16** | 6.063 | 0.7004 | **4.148** | 3.551 | 130.6 |
| | VARestorer-1 | 22.78 | 0.6453 | 0.3249 | **0.5655** | **71.37** | **4.763** | **0.7423** | **4.601** | **4.180** | **117.2** |

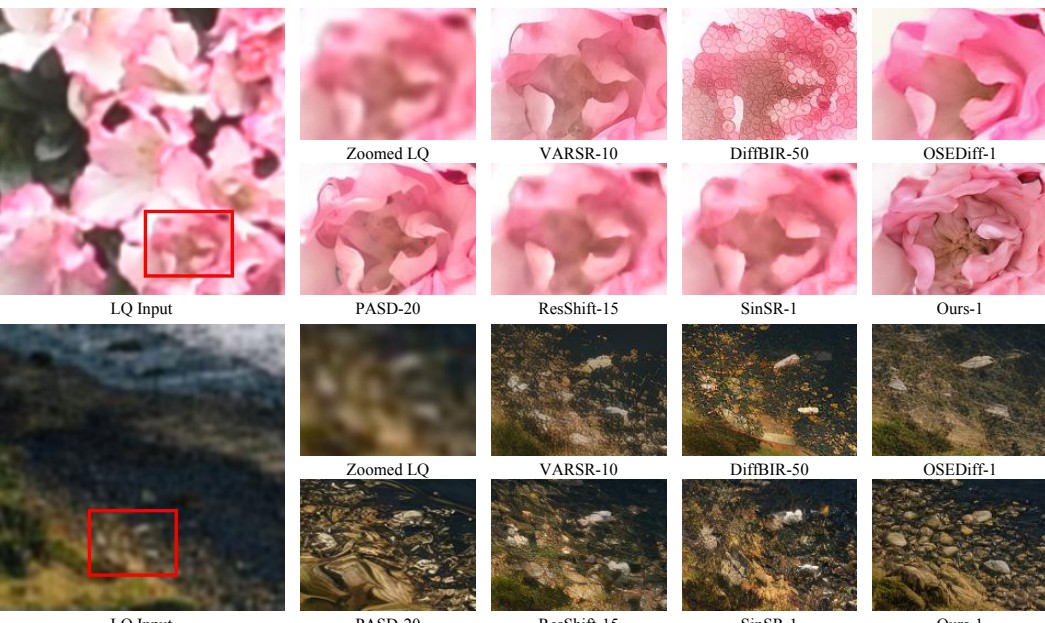

Figure 4: **Qualitative comparisons on real-world datasets.** Our VARestorer delivers exceptional details with just one-step inference. The numbers following each method indicate the corresponding inference steps. Please zoom in for a better view.

have their limitations in assessing visual quality as they often penalize high-frequence details in our generated images, *e.g.*, hair texture. Therefore, we also include the widely-used FID (Heusel et al., 2017) score to provide a distribution-level evaluation of overall image quality and realism. Additionally, we leverage six widely adopted non-reference metrics, including MANIQA (Yang et al., 2022), MUSIQ (Ke et al., 2021), CLIPIQA (Wang et al., 2023a), LIQE, NIQE and QALIGN, to assess image quality. Finally, in order to highlight the practicality of VARestorer, we also compare the inference time of our framework with other competing approaches.

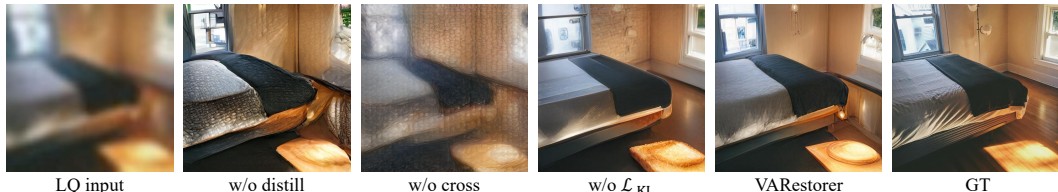

| LQ input | w/o distill | w/o cross | w/o $\mathcal{L}_{KL}$ | VARestorer | GT |

Figure 5: **Visual results of the ablations.** Our distillation method, cross-scale attention, and distribution matching collectively enhance the visual quality of the generated images by reducing artifacts, preserving fine details, and ensuring better structural consistency.

## 4.2 MAIN RESULTS

**Quantitative Comparisons**. As shown in Table 1, we present a comprehensive comparison with DiffBIR (Lin et al., 2023), SeeSR (Wu et al., 2024b), PASD (Yang et al., 2024), ResShift (Yue et al., 2023), VARSR (Qu et al., 2025), OSEDiff (Wu et al., 2024a) and SinSR (Wang et al., 2024) on the three benchmark datasets (DIV2K-Val, DrealSR, and RealSR). Our method, VARestorer, achieves consistently strong performance across various metrics, particularly in the no-reference perceptual metric. On all three datasets, VARestorer attains the highest CLIPIQA and NIQE scores, as well as top-2 LIQE and MUSIQ scores, indicating that our restorations are both perceptually appealing and well-aligned with aesthetic preferences. Although some methods (e.g., PASD and OSEDiff) perform well on certain reference quality metrics (such as PSNR or LPIPS), their advantages often come at the expense of either lower visual quality or more inference steps. In contrast, VARestorer strikes a more favorable balance, delivering top-tier perceptual quality and aesthetics while maintaining competitive fidelity—exemplified by our best FID on RealSR. Parameter and inference efficiency (Table 2) further reinforce our advantage: despite using fewer trainable parameters and faster inference than Qu et al. (2025), VARestorer delivers superior performance. These results underscore the effectiveness of our design, which unifies high perceptual quality, fidelity, efficiency, and distribution alignment into a single, efficient framework.

**Qualitative Comparisons**. Figure 4 presents a visual comparison with several representative methods on two challenging real-world image restoration cases. In the first row, most methods struggle with preserving the intricate petal structure—some introduce blurriness (e.g., PASD and SinSR), others misinterpret colors (e.g., OSEDiff), or generate unrealistic textures (e.g., DiffBIR). While certain methods, such as PASD, retain the overall flower shape and color, they still struggle with generating convincing petal details. In contrast, our approach effectively restores sharp and natural petal structures, accurately capturing subtle folds and smooth color transitions. A similar conclusion can be drawn from the second row. While other methods over-smooth textures or introduce artifacts near transitions (rocks and water), our method yields more authentic textures and clearer boundaries with fewer artifacts. Such fidelity and detail preservation highlight the effectiveness of our design in leveraging powerful image priors while maintaining robust semantic consistency. Overall, VARestorer consistently produces high-quality, visually appealing results, confirming its superiority over competing methods. Additional comparisons are provided in Section B.

## 4.3 ANALYSIS

In this section, we will conduct detailed ablation experiments on DIV2K-Val dataset to further evaluate the effectiveness of each of the components in VARestorer. We further extend VARestorer to tasks such as deraining and low-light enhancement, with details provided in Section C.

**VAR compression in one step.** The VAR achieves high-quality outcomes with a multi-step next-scale prediction approach. We experiment with several model structures using the next-scale prediction approach—e.g., ControlNet, adapter, or directly concatenating them with tokens—to inject low-quality image features. However, these attempts yielded worse restoration results, with pronounced artifacts and blurriness. The primary reason lies in the error propagation intrinsic to next-scale prediction: a subtle mistake made in the initial token predictions is carried over to subsequent scales, accumulating at each step. Unlike image generation, which can tolerate some deviations, restoration requires a high degree of alignment between the output and the input image, making such error accumulation particularly detrimental. To mitigate this issue, we distilled VAR into a one-step model

Table 2: **Parameter and inference analysis.** Our generator demonstrates a balance between computational efficiency and performance.

| Method | Trainable Params. | Inference Time (s) | MANIQA↑ | MUSIQ↑ |
|---|---|---|---|---|
| DiffBIR | 380.0M | 10.27 | 0.5664 | 69.87 |
| SeeSR | 749.9M | 7.18 | 0.5036 | 68.67 |
| PASD | 625.0M | 4.58 | 0.4371 | 67.78 |
| ResShift | 118.6M | 1.13 | 0.3693 | 58.90 |
| OSEDiff | 8.5M | 0.18 | 0.4410 | 67.96 |
| VARSR | 1101.9M | 0.63 | 0.5173 | 71.48 |
| VARestorer | 27.3M | 0.23 | **0.5590** | **72.32** |

Table 3: **Ablation studies.** We evaluate VARestorer's components and demonstrate that it outperforms multi-step VAR and scale-wise causal attention. Additionally, distribution matching further improves image quality and consistency.

| Method | LPIPS↓ | MUSIQ↑ | NIQE↓ | CLIPIQA↑ |
|---|---|---|---|---|
| w/o distill | 0.3723 | 62.22 | 6.283 | 0.4794 |
| w/o cross | 0.4224 | 63.72 | 6.029 | 0.3910 |
| w/o $\mathcal{L}_{KL}$ | 0.3214 | 69.73 | **4.372** | 0.6682 |
| VARestorer | **0.3131** | **72.32** | 4.410 | **0.7669** |

that generates tokens for all scales in a single inference step. Although this inference strategy deviates from the VAR's training procedure, the resulting images remain well-aligned with the input while preserving high visual quality. We report the performance of multi-step VAR with ControlNet in Table 3 and Figure 5 (w/o distill). While promising in diffusion-based models, this approach produces unsatisfactory results with noticeable artifacts. Moreover, compared with VAR-based Qu et al. (2025) (10 steps, >1B parameters) in Table 2, our VARestorer delivers superior results in just one step. These findings demonstrate the effectiveness of our method.

**Enhanced distillation via distribution matching.** In the training phase, relying solely on LPIPS and MSE losses between the generated and ground-truth images effectively constrains the model to learn a one-to-one mapping from the low-quality input to a single high-quality output. However, this approach conflicts with the inherent one-to-many nature of image restoration, substantially increasing training difficulty and limiting the diversity of generated images. To address this issue, we introduce a distribution matching strategy that leverages a pre-trained VAR model as the teacher. Specifically, we incorporate the KL divergence between the probability distributions predicted by our model and those from the teacher into the training loss. By integrating the generative capabilities of the I2V model, we transition from a strict one-to-one training paradigm to a distribution-matching approach, thereby accelerating training and improving the visual quality of the generated outputs. As shown in Table 3 and Figure 5, we conduct an ablation study by removing this design (w/o $\mathcal{L}_{KL}$). The student model can still give clean images but lack realness and some unrealistic texture occurs. This underscores the crucial role of $\mathcal{L}_{KL}$ in enhancing image quality and preserving realistic details.

**Cross-scale conditioning for high quality.** Due to the sequential generation nature of the VAR model, low-level scales lack direct access to information from the higher-level scales. This absence of cross-scale interaction leads to unawareness of the low-level token maps about how to construct a proper foundation map for later scales, making it difficult to reconstruct high-quality images during the ISR process. To address this, we replace scale-wise causal attention in the autoregressive transformer with full attention, allowing information to flow across different scales. To assess the impact of this change, we conduct an ablation experiment where we maintain the original causal attention structure (w/o cross). As shown in Table 3 and Figure 5, our results demonstrate that cross-scale conditioning can bring significant improvement in both quantitative metrics and visual quality. The causal attention captures the relationship between different resolution levels, resulting in global blurring and patch-like artifacts. We hypothesize this issue arises due to the disharmony within token maps across scales, further highlighting the importance of enabling cross-scale communication.

**Limitations.** VARestorer delivers high-quality, one-step restoration but may struggle with severe noise or heavy compression. Some failure cases are shown in Section E.

## 5 CONCLUSION

We propose VARestorer, a one-step real-world image super-resolution framework that achieves both efficiency and effectiveness. By distilling a pre-trained VAR model with distribution matching and integrating cross-scale pyramid conditioning, VARestorer effectively aligns the generated token distributions with the pre-trained VAR, ensuring high-fidelity restoration with minimal computational cost. This approach mitigates error accumulation in autoregressive models while fully leveraging VAR's generative priors. We hope our work inspires future research to further explore efficient generative priors for image restoration and enhancement.

ACKNOWLEDGMENTS

This work was supported in part by the National Natural Science Foundation of China under Grant 62125603, Grant 62336004, Grant 62321005, and in part by the Beijing Natural Science Foundation under Grant No. L247009.

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

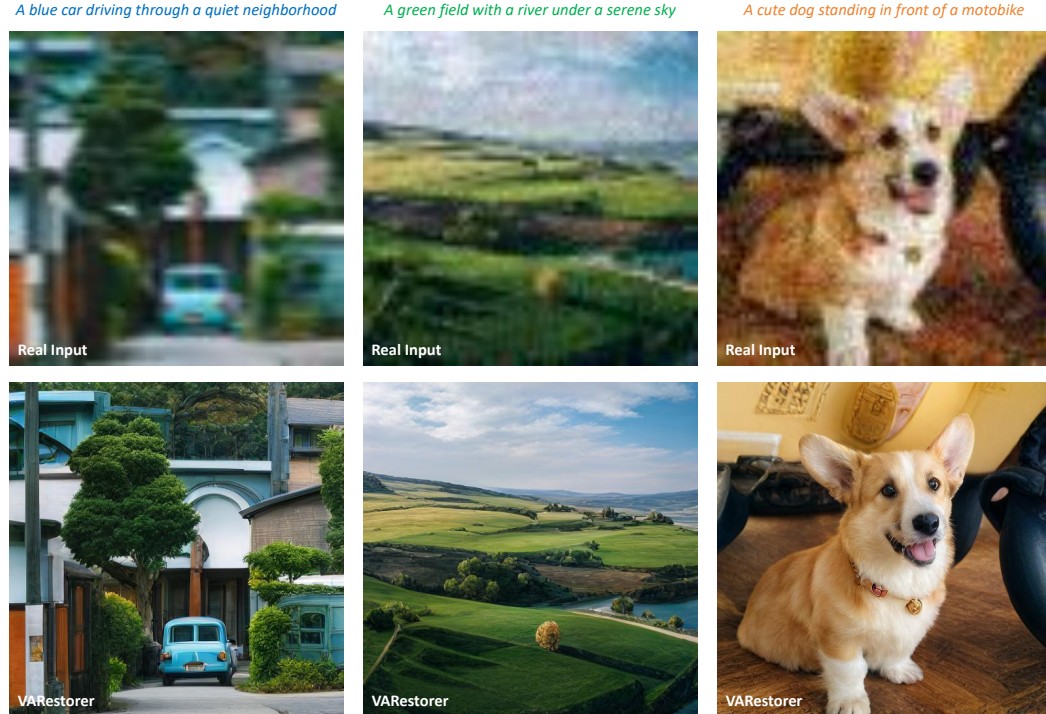

*A blue car driving through a quiet neighborhood*  *A green field with a river under a serene sky*  *A cute dog standing in front of a motobike*

Figure A: VARestorer achieves strong one-step restoration by effectively leveraging the knowledge of the pre-trained VAR model.

## A   DETAILED IMPLEMENTATIONS

We employ the widely used Visual Autoregressive (VAR) model, Infinity-2B (Han et al., 2024), which has 32 transformer layers, an embedding dimension of 2048, and 16 attention heads to leverage its generative capabilities for image restoration. The training process is conducted on the LS-DIR (Li et al., 2023) dataset, which consists of approximately 8,5000 images. All images are resized to a resolution of $512 \times 512$, and we utilize BLIP (Li et al., 2022) to generate corresponding image captions. To mitigate error accumulation caused by next-scale prediction, we distill the pretrained model into a one-step model. Specifically, we concatenate the input image tokens from all scales and predict the output in a single step. Cross-scale attention is incorporated to ensure that all input information contributes to the prediction of each token. To preserve the visual priors of the pretrained model, we integrate LoRA parameters with a rank of four into the transformer while freezing all other parameters. The trainable parameters amount to approximately 27.3M, accounting for only 1.2% of the total model parameters. We employ a distribution matching method to effectively incorporate the generative capabilities of the I2V model and prevent one-to-one mapping. Since the Infinity model is originally designed to generate images at a resolution of $1024 \times 1024$, we fine-tune it on the $512 \times 512$ dataset and use the fine-tuned model as the teacher. The KL divergence between the probability distributions predicted by our model and those from the teacher is computed and used as a loss term during training. The training process is conducted on 8 NVIDIA LS20 GPUs, each equipped with 48GB of VRAM. With just a single epoch of training on the LSDIR dataset, the model attains strong restoration capabilities, requiring approximately 4.5 hours. During inference, the model processes each image in an average of 0.33 seconds, making it comparable with GAN-based and diffusion distillation methods. Figure A demonstrates the model's ability to effectively restore high-quality images from degraded inputs.

**Details of the cross-scale attnetion.** Our cross-scale full attention is a deliberate architectural deviation from the original VAR's block-wise causal attention. In the standard VAR design, each scale attends only to past scales through a block-wise causal mask, enforcing an autoregressive dependency structure. While this preserves strict generative ordering for upsampling, it also limits feature interaction across scales, often resulting in suboptimal restoration quality due to insufficient multi-scale information flow. In VARestorer, we instead employ a full attention mask across all

scales (illustrated in Figure 3), allowing features from every scale to interact bidirectionally. This modification is crucial for image restoration: unlike generation from tokens, ISR benefits from rich cross-scale fusion, where both coarse structures and fine textures reinforce each other. Importantly, although this departs from VAR's autoregressive mechanism, we find that: (1) The modification preserves most of the pre-trained VAR representational capacity, as validated by our strong quantitative and qualitative results (Figure 5 and Table 3). (2) It significantly improves restoration quality, addressing the limitations caused by causal masking and enabling stronger multi-scale consistency. Overall, our design prioritizes effective feature integration over strict autoregressive ordering, which is more suitable for the one-step restoration setting.

## B    More Qualitative Comparisons

We provide additional qualitative results to further demonstrate the effectiveness and versatility of our framework. As shown in Figure B, our method produces highly realistic, high-quality outputs that closely match the input image. In the first example, for instance, AddSR and DiffBIR struggle to restore texture, resulting in oversmoothed or artifact-laden details. OSEDiff recovers some structural information but lacks high-frequency details and intricate patterns. PASD and ResShift tend to generate either unnatural patterns or blurry edges, while SinSR fails to adequately restore the boundary regions. In contrast, our approach effectively preserves subtle textures and structural fidelity, yielding a more realistic and visually coherent outcome. This highlights the strength of our framework in capturing both global structure and fine-grained details.

## C    Generalization to More Tasks

To assess generalization, we apply our VARestorer to several challenging super-resolution (SR) scenarios that deviate from the training degradations. Below, we provide representative visual examples in the supplement and summarize results for four special cases.

We apply our model to deraining and low-light enhancement with slight fine-tuning. As shown in Figure C, our VARestorer consistently produces satisfactory results.

We additionally test on RealBlur-SR, heavy JPEG compression, and samples from social media. As shown in Figure D, our model performs consistently well across these challenging scenarios.

Generalization to unseen degradations is challenging. However, recent advances in large-scale pre-training allow generative models (e.g., diffusion, VAR) to encode robust priors across diverse scenarios. By distilling from such models, we benefit from this generality. We demonstrate this on *salt-and-pepper noise*, which is unseen during training and structurally different from our degradation pipeline. As shown in Figure E, our model produces strong results.

Our model uses fixed resolution during evaluation to align with standard practice in prior SR works (e.g., DiffBIR, OSEDiff, AddSR). These methods all operate on a fixed image size during inference. However, our model can indeed handle *diverse resolutions*: (1) The pretrained VQVAE (default size 1024) can effectively encode and reconstruct smaller images (e.g., 512, 768). (2) During inference, we first resize the LR inputs to $512^2$ like OSEDiff, enabling arbitrary lower-resolution inputs. (3) For higher resolutions, we can adopt two approaches: *tiling-based inference* like diffusion-based methods and *fine-tuning* on larger and mixed scales, both supported by the teacher (default size 1024), as shown in Figure F.

## D    More discussions

### D.1    Ablation on textual prompts

To evaluate the contribution of textual conditioning, we conduct an ablation study by removing the prompt at inference time and feeding an empty text input ("w/o prompt"). All other settings remain unchanged. Across benchmarks, the absence of prompt guidance leads to noticeable performance drops, particularly in semantic consistency and fine-grained structure recovery. Without textual cues, the model tends to produce overly smooth textures and occasionally drifts toward ambiguous object shapes. In contrast, using the prompt provides high-level contextual signals that help

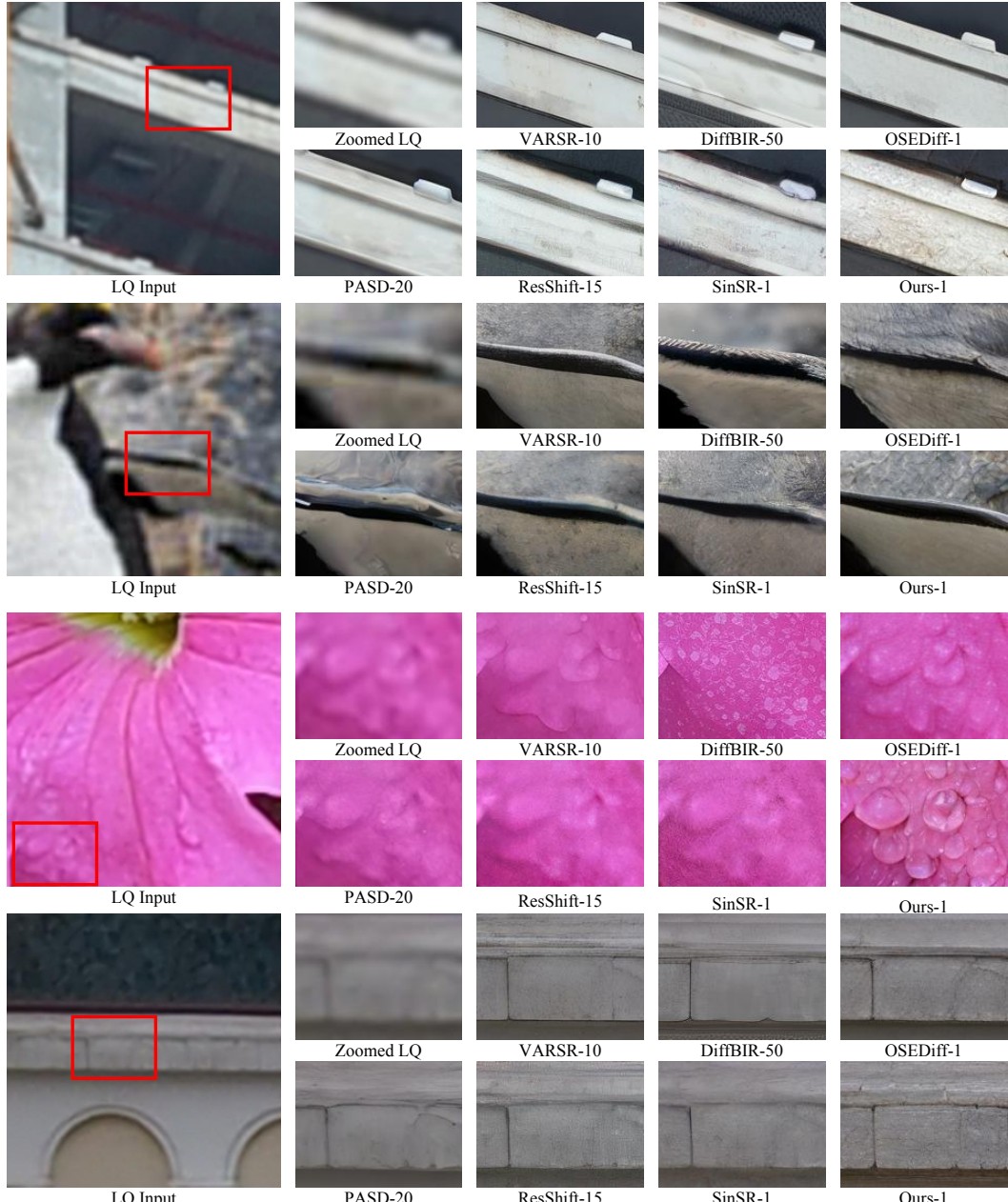

Figure B: **More qualitative comparisons.** Our VARestorer delivers exceptional details with just one-step inference. The numbers following each method indicate the corresponding inference steps.

the model stabilize its predictions, preserve object identities, and generate sharper details. These results demonstrate that the textual prompt is not merely auxiliary, but provides meaningful semantic constraints that improve both fidelity and consistency in the reconstructed images. We include quantitative comparisons and visual examples in Table A.

## D.2 FULL TUNING VS. LORA TUNING

We further compare full fine-tuning with parameter-efficient LoRA tuning under the same training setup in Table A. Interestingly, full tuning yields slightly lower quantitative metrics and requires a substantially longer time to converge. We attribute this behavior to the fact that updating all parameters can disturb the pre-trained generative prior, making optimization less stable and occasionally leading to degraded image quality. In contrast, LoRA tuning preserves most of the pretrained weights and introduces only lightweight, task-specific adaptations. This allows the model to retain

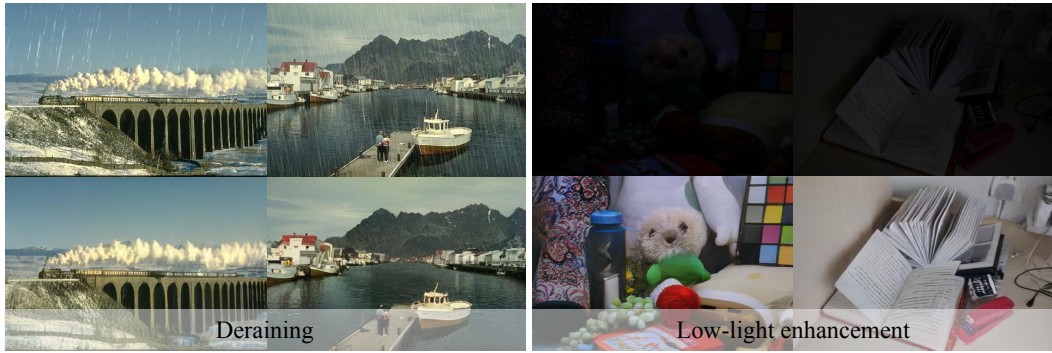

Figure C: **More tasks supported by VARestorer.** VARestorer supports various tasks, including deraining and low-light enhancement with slight fine-tuning. Please zoom in for a better view.

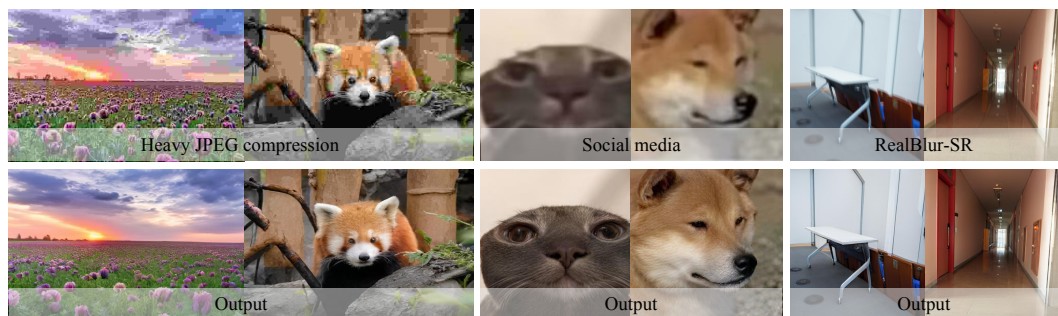

Figure D: **JPEG artifact removal.** VARestorer can effectively remove JPEG artifacts and recover clear textures. Please zoom in for a better view.

its generative capability while efficiently learning ISR-specific behaviors, resulting in faster convergence and better performance.

### D.3 $1024 \times 1024$ RESOLUTION.

We also study the effect of using 1024×1024 training resolution. Although the base VAR model is pretrained to generate 1024px images, our experiments show that 1024px training yields competitive results but does not provide notable performance improvements for ISR. (Table A). However, it introduces a substantial increase in training time and computational cost (Figure H, left, Ours-1024). Interestingly, the pre-trained model already exhibits strong generation capability at smaller resolutions (e.g., 512 and 768). We attribute this to the multi-scale VAE design, which provides stable and high-quality representations across resolutions. As a result, training at lower resolutions achieves comparable quality while being far more efficient.

### D.4 TRAINING SUFFICIENCY

VARestorer fine-tunes only 1.2% of the pre-trained VAR model, so it does not require long training schedules to reach convergence. This behavior is consistent with prior tuning-based ISR approaches. For example, OSEDiff (Wu et al., 2024a) fine-tunes diffusion models using a batch size of 16 for 20K steps (∼1 day on 4×A100). In comparison, our setup (8×L20, batch size 32) reaches convergence within 10K steps (∼2 days, 3.7 epochs). To validate this, we include a training MSE loss curve in Figure H (left, Ours-512), which clearly shows that the loss stabilizes well before 10K steps. We also trained variants for 20K and 25K steps and observed no meaningful improvement across any metric. This confirms that the model has already converged under our standard schedule and that additional training provides negligible benefit. These results demonstrate that the proposed

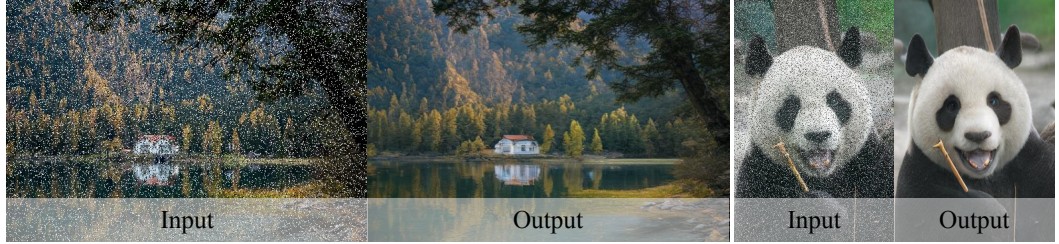

Figure E: **Salt & Pepper noise removal.** VARestorer can effectively remove salt & pepper noise and recover clear textures. Please zoom in for a better view.

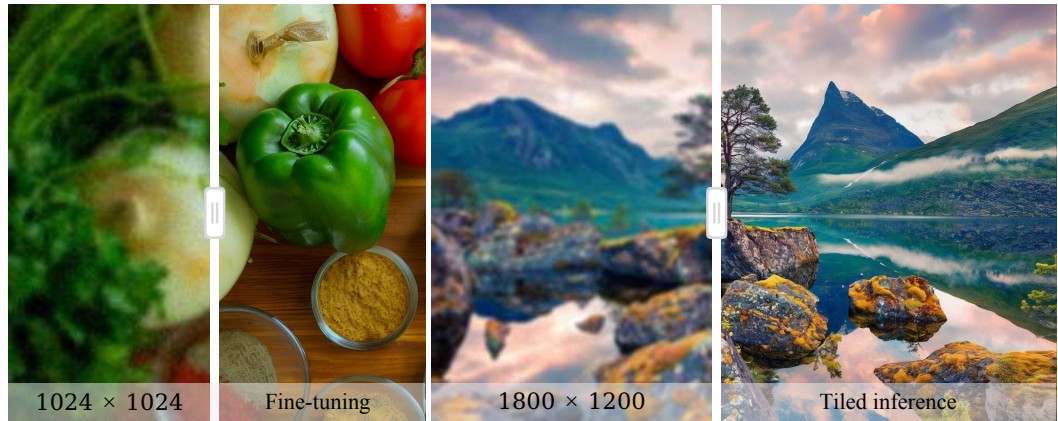

Figure F: **High-resolution image restoration.** VARestorer can effectively restore high-resolution images (e.g., 1024×1024) with high-quality details. Please zoom in for a better view.

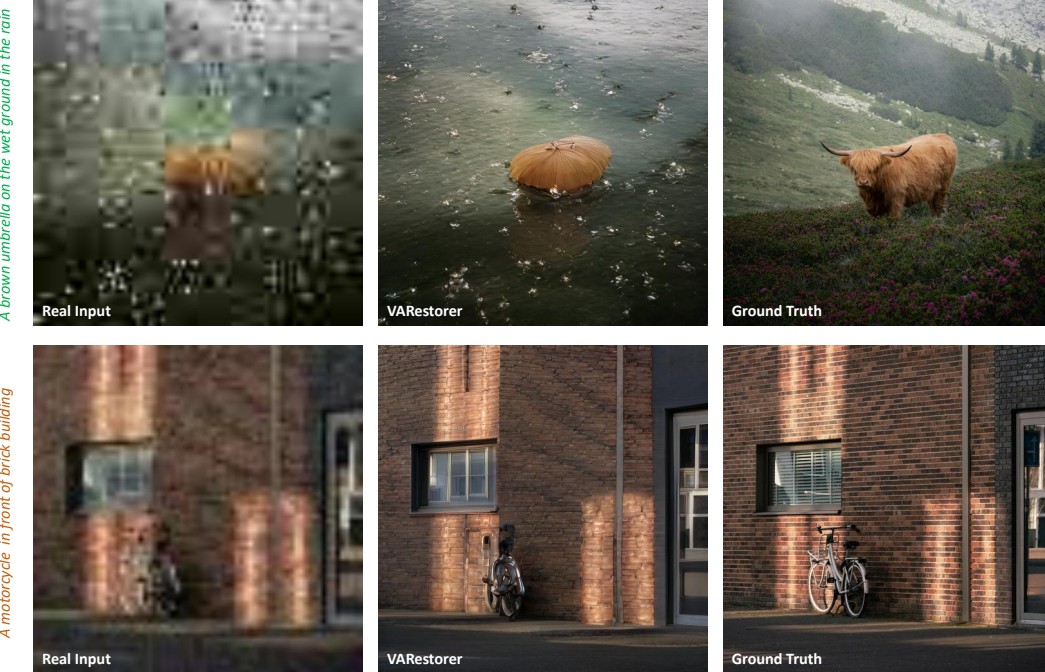

Figure G: **Failure cases.** VARestorer may struggle with certain extremely challenging tasks, such as severe degradation or complex noise patterns. Please zoom in for a better view.

lightweight fine-tuning procedure is sufficient and efficient for adapting the VAR backbone to the ISR task.

## D.5 COMPUTATION ANALYSIS

We report the FLOPs of VARestorer and competing methods on $512 \times 512$ images in Table B. Since FLOPs vary slightly with textual prompt length, we estimate the computational cost by averaging

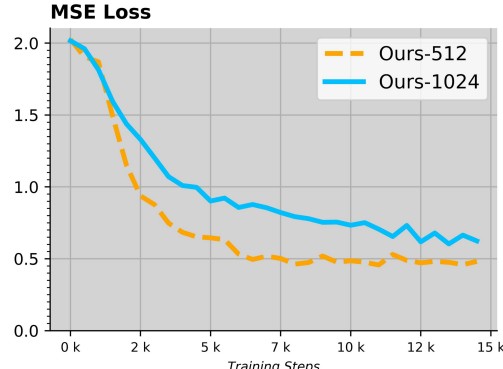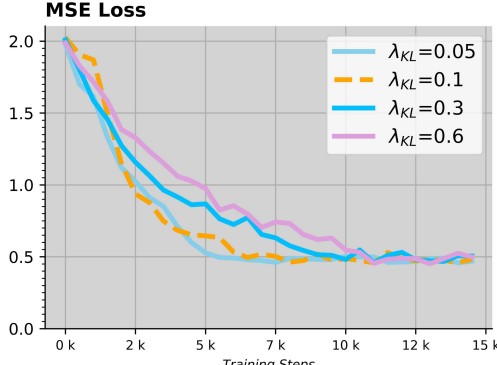

Figure H: **Loss curves of the ablations.** We present the MSE loss curves for key ablation settings, including training with 1024px inputs and sensitivity analysis of the super-parameter $\lambda_{\text{KL}}$.

Table A: **Additional Ablation Studies.** We conduct ablations to evaluate the impact of textual prompts, tuning strategies, and training image resolution on VARestorer's performance.

| Method | LPIPS↓ | MUSIQ↑ | NIQE↓ | CLIPIQA↑ |
|---|---|---|---|---|
| w/o prompt | 0.3201 | 67.05 | 4.831 | 0.7332 |
| w/o restorer | 0.3150 | 72.27 | 4.458 | 0.7624 |
| full tuning | 0.3127 | 70.58 | 4.976 | 0.6864 |
| 1024 resolution | **0.3082** | 72.11 | 4.427 | 0.7645 |
| VARestorer | 0.3131 | **72.32** | **4.410** | **0.7669** |

over 10 randomly sampled prompts. Our method requires only 1536 GFLOPs, which is roughly 10% of diffusion-based approaches such as SeeSR (Wu et al., 2024b) and PASD (Yang et al., 2024), both of which rely on multi-step sampling. Despite this significant reduction in computation, VARestorer still achieves state-of-the-art performance on perceptual metrics, including MANIQA and MUSIQ, demonstrating a strong balance between efficiency and image quality. We further analyze the computational overhead of the cross-scale pyramid conditioning: compared with the original block-wise causal attention (w/o cross in Table 3, ∼1200 GFLOPs for $512 \times 512$ images), full cross-scale attention adds roughly 330 GFLOPs, resulting in ∼1530 GFLOPs. This moderate increase is justified by the notable gains in perceptual metrics and visual fidelity.

## D.6 DISCUSSION ABOUT METRICS

Traditional full-reference metrics such as PSNR and SSIM have long been used to evaluate image restoration performance. While these metrics measure pixel-wise similarity or structural consistency, they do not fully capture human perceptual preference. In particular, they often favor overly smooth or blurry reconstructions that minimize low-level errors, even if such outputs lack realistic textures and high-frequency details. This limitation has been extensively discussed in prior works (Yu et al., 2024b; Blau & Michaeli, 2018; Jinjin et al., 2020; Gu et al., 2022; Liang et al., 2021; Lin et al., 2023).

Recent generative restoration methods, including VARestorer, produce richer, visually appealing details that align better with human perception (Figure 4 and Figure B). As a result, such methods may score lower on PSNR or SSIM (Table 1) despite providing outputs that are sharper, more natural, and more faithful to real-world image distributions. To provide a more comprehensive evaluation, we also report non-reference perceptual metrics such as MANIQA and CLIPIQA, where VARestorer achieves substantial improvements (Table 1).

A common concern is whether the improvement in perceptual metrics arises from artificial high-frequency patterns rather than genuine detail recovery. In our case, the gains do not stem from hallucinated noise. As shown in our qualitative analyses (Figure I), the model reliably reconstructs both high-frequency details (e.g., flowers) and low-frequency structures (e.g., desert, sea), demonstrating balanced reconstruction across frequency components. However, in extremely degraded cases where the input is too blurry to reveal its original content, perfect fidelity is fundamentally

Table B: **Computation and performance analysis.** Our one-step generator achieves an effective trade-off between computational efficiency and restoration performance.

| Method | Trainable Params. | Inference Time (s) | GFLOPs | MANIQA↑ | MUSIQ↑ |
|---|---|---|---|---|---|
| DiffBIR | 380.0M | 10.27 | 12117 | 0.5664 | 69.87 |
| SeeSR | 749.9M | 7.18 | 32928 | 0.5036 | 68.67 |
| PASD | 625.0M | 4.58 | 14562 | 0.4371 | 67.78 |
| ResShift | 118.6M | 1.13 | 2745 | 0.3693 | 58.90 |
| OSEDiff | 8.5M | 0.18 | 1133 | 0.4410 | 67.96 |
| VARestorer | 27.3M | 0.23 | 1536 | **0.5590** | **72.32** |

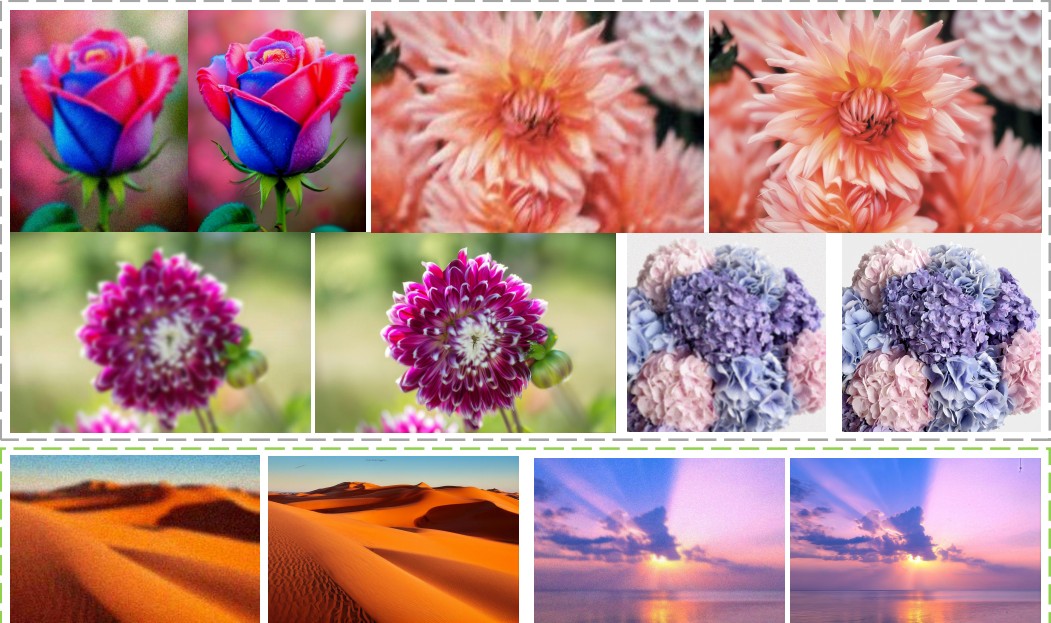

Figure I: **Qualitative results across frequency components.** VARestorer generates visually pleasing outputs, preserving both high-frequency details (top: flowers) and low-frequency structures (bottom: desert and sea). Each example shows the low-quality input on the left and our restored result on the right.

unattainable. In these scenarios, the model leverages its generative capability to produce plausible, natural-looking structures that align with real-world image statistics. While this may lead to results that are slightly less faithful to the exact ground truth, they are visually coherent and far closer to the natural image distribution, avoiding the overly smooth and unusable outputs typical of PSNR-oriented methods. We argue that such perceptual reconstruction—producing realistic textures when the signal is insufficient—is more aligned with real-world practical needs than strictly preserving low-level metrics at the cost of visual quality.

To clarify the LPIPS behavior of generative-prior SR models, we include a comparison with BSR-GAN (Zhang et al., 2021a) (Table C, Figure J). Under heavy degradation, BSRGAN attains higher fidelity scores (PSNR/SSIM) mainly because its outputs become overly smooth and blurry, reducing feature-space distance to the ground truth. In contrast, our method restores natural high-frequency textures and sharper structures, which improves perceptual quality but can increase LPIPS when the true details are unrecoverable. Importantly, this behavior is a general feature of generative models rather than arbitrary hallucination: under light degradation our model shows no semantic drift, and under heavy degradation all generative-prior approaches necessarily rely on learned distributions. The resulting textures may deviate from the exact GT pixels but remain plausible, statistically natural, and structurally consistent with the input. As shown in Figure J, the LPIPS increase on DRealSR/RealSR mainly results from realistic high-frequency variations (e.g., foliage, brick patterns, window textures) rather than incorrect semantic content.

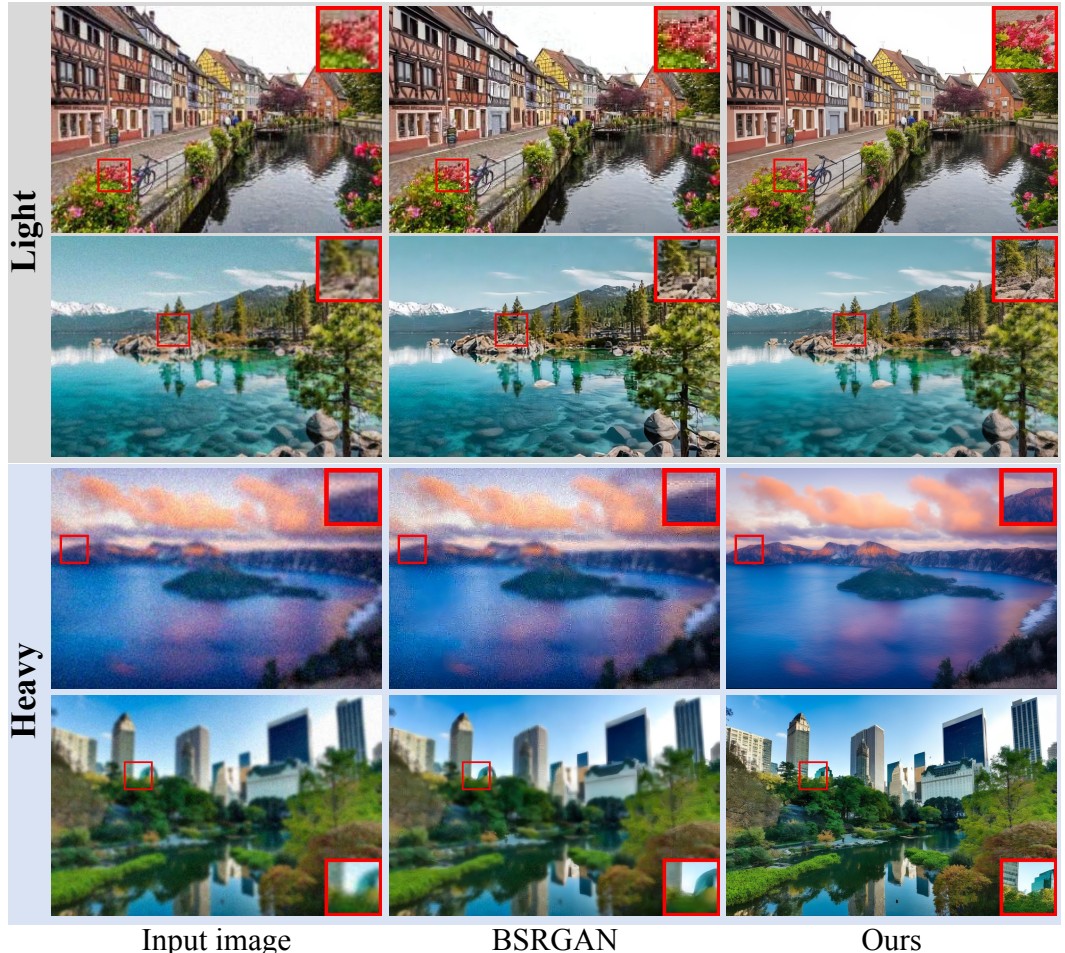

Figure J: **Qualitative comparison with BSRGAN** (Zhang et al., 2021a) under light (top) and heavy (bottom) degradation. We show cases under both light (top) and heavy degradation (bottom). BSR-GAN fails under severe degradation, while our method produces plausible restorations.

Table C: **Quantitative comparison with BSRGAN** (Zhang et al., 2021a). VARestorer achieves substantially higher perceptual quality metrics.

| Datasets | Methods | PSNR↑ | SSIM↑ | LPIPS↓ | MANIQA↑ | MUSIQ↑ | NIQE↓ | CLIPIQA↑ | FID↓ |
|----------|---------|-------|-------|--------|---------|--------|-------|----------|------|
| DIV2K-Val | BSRGAN | **24.58** | **0.6269** | 0.3351 | 0.5071 | 61.20 | 4.7518 | 0.5247 | 44.23 |
| | VARestorer-1 | 21.08 | 0.5355 | **0.3131** | **0.5590** | **72.32** | **4.410** | **0.7669** | **31.11** |
| DrealSR | BSRGAN | **28.75** | **0.8031** | **0.2883** | 0.4878 | 57.14 | 6.5192 | 0.4915 | 155.63 |
| | VARestorer-1 | 24.31 | 0.6894 | 0.3584 | **0.5638** | **69.49** | **5.494** | **0.7810** | **149.7** |
| RealSR | BSRGAN | **26.39** | **0.7654** | **0.2670** | 0.5399 | 63.21 | 5.6567 | 0.5001 | 141.28 |
| | VARestorer-1 | 22.78 | 0.6453 | 0.3249 | **0.5655** | **71.37** | **4.763** | **0.7423** | **117.2** |

## D.7 SENSITIVITY ANALYSIS OF $\lambda$S

We conduct a sensitivity analysis to evaluate how the choice of loss weights affects training stability and final performance. In VARestorer, the total training loss combines KL divergence ($\mathcal{L}_{KL}$), MSE loss ($\mathcal{L}_{MSE}$), and perceptual loss ($\mathcal{L}_{perc}$), weighted by hyperparameters $\lambda_{KL}$, $\lambda_{MSE}$, and $\lambda_{perc}$. We vary $\lambda_{KL}$ in $\{0.05, 0.1, 0.3, 0.6\}$ while keeping the other weights fixed. As shown in Figure H (right), larger $\lambda_{KL}$ tends to slightly slow down convergence, but all configurations eventually reach similar MSE loss levels, indicating that the final performance is robust to moderate changes in $\lambda_{KL}$. A broader hyperparameter search further confirms that the final restoration quality is relatively insensitive as long as the weights are within reasonable ranges: $\lambda_{KL} \in [0.05, 0.8]$, $\lambda_{MSE} \in [0.3, 1]$,

and $\lambda_{\mathrm{perc}} \in [0.2, 1]$. The optimal combination selected in our experiments is $\lambda_{\mathrm{KL}} = 0.1$, $\lambda_{\mathrm{MSE}} = 0.5$, and $\lambda_{\mathrm{perc}} = 0.25$, which balances convergence speed and restoration quality.

### D.8 EFFECT OF SWINIR PREPROCESSING

We conduct an ablation to assess the impact of the SwinIR preprocessor on VARestorer's performance (w/o restorer). The preprocessor is applied to coarsely remove simple degradations (e.g., noise) before extracting textual prompts, allowing the generative stage to focus on high-frequency details.

Our experiments in Table A show that while SwinIR improves the accuracy of prompt extraction and slightly enhances restoration quality, the absence of SwinIR results in only a modest performance drop across evaluated metrics. Importantly, even without the preprocessor, VARestorer still surpasses prior methods on most quantitative metrics, indicating that the generative framework itself is robust.

## E FAILURE CASES

While our framework demonstrates impressive performance across a wide range of image types, it exhibits certain limitations when handling particularly complex tasks. As illustrated in Figure G, we present two representative failure cases that highlight these challenges. In the first instance, the input image suffers from severe degradation and substantial noise contamination. This condition leads to an entirely incorrect prompt from the caption model. Consequently, due to both the low-quality input and the erroneous prompt, VARestorer generates an image that significantly deviates from the ground truth, thereby revealing its limitations in restoring images with substantial noise interference.

The second case presents a scenario where the bicycle's skeletal structure, particularly the handlebars, appears relatively slender against the brick building background. The noise interference disrupts the bicycle's structural integrity during the degradation process, making it challenging to recognize. While VARestorer successfully restores other components of the image, it fails to reconstruct the complete bicycle structure.

These observations indicate that the limitations under severe degradation stem from two main factors: (1) insufficient informative content in the input, which reduces the ability of the model to condition its generation accurately, and (2) the inherent one-step inference and generative mechanism, which, while efficient, cannot fully compensate for missing or ambiguous signals. In such scenarios, VARestorer relies on learned priors to generate plausible structures, which may result in outputs that deviate from the original content while still being realistic.

Although VARestorer consistently produces high-quality and coherent results in most restoration tasks, it encounters difficulties with certain exceptionally challenging scenarios. These observations underscore the need for further refinement of the framework to enhance its robustness in handling complex and noisy image restoration tasks.

