# OpenReview forum: "VARestorer: One-Step VAR Distillation for Real-World Image Super-Resolution"
_ICLR.cc/2026/Conference — ICLR 2026 Poster_

### Official Review · Reviewer_gJQV · 2025-10-29

**Soundness:** 3
**Presentation:** 2
**Contribution:** 2
**Rating:** 6
**Confidence:** 3

**Summary:**

This paper proposes VARestorer, a one-step super-resolution framework that addresses error accumulation in visual autoregressive models through distribution matching distillation. The method incorporates cross-scale pyramid conditioning to enable multi-scale information interaction while maintaining the original model's generative capabilities. Experiments demonstrate superior performance, achieving an optimal balance between efficiency and restoration quality.

**Strengths:**

1. The paper adapts VAR models to the real-world image super-resolution (Real-ISR) task. The proposed distillation approach directly tackles the critical issue of error accumulation inherent in autoregressive models for restoration.
2. The pyramid image conditioning with cross-scale attention enables bidirectional scale-wise interactions and fully utilizes the information of the input image while adapting to the autoregressive mechanism.

**Weaknesses:**

1. The computational cost of the cross-scale pyramid conditioning should be discussed, with quantitative analysis to justify its overhead in relation to performance gains.
2. The labels and arrows in Figure 3 are unclear. Please define $\mathcal{D}$ and $\mathcal{E}$ (possibly Encoder and Decoder) in the caption or figure, and redraw the arrows to unambiguously show the process flow and component interactions.
3. The paper employs BLIP to generate textual prompts, yet its necessity remains unverified. The claim that text guidance improves restoration quality lacks support from ablation studies.
4. Although the high efficiency of training only 1.2% of the parameters was emphasized, no performance comparison was made with full parameter fine-tuning. It is suggested to add the baseline experimental results of full parameter fine-tuning to conclusively prove that LoRA can balance efficiency while maintaining performance.
5. The computational efficiency claims lack comprehensive quantification. While inference time is reported, metrics like FLOPs or MACs are missing, making cross-architecture comparison difficult.

**Questions:**

1. How does the distillation process prevent the student model from learning and reproducing any inherent errors or suboptimal predictions made by the teacher model during distribution matching?
2. By transforming a sequential decision process into a one-shot mapping, the model might sacrifice some expressive power for computational efficiency. Could you discuss the theoretical limitations this imposes, especially when addressing highly complex or unseen degradations?

---

> ### Author Response · Authors · 2025-11-22
>
> We sincerely thank Reviewer gJQV for the valuable feedback. We address your concerns and questions below.
>
> > ### **About computation cost**
>
> **[Reply]:** Thank you for the comment. We have analyzed the computational overhead of the cross-scale pyramid conditioning. Compared with the original scale-wise causal attention (~1200 GFLOPs for $512\times512$ images), the full cross-scale attention **increases the cost by roughly 330 GFLOPs**, resulting in ~1530 GFLOPs. Despite this moderate increase, the performance gains—especially in perceptual metrics and visual fidelity—justify the additional computation. We have added this discussion and quantitative analysis in the revision, included in **Section D.5**, alongside a comparison with the FLOPs of other methods：
> | Method       | Trainable Params. | Inference Time (s) | GFLOPs | MANIQA ↑ | MUSIQ ↑ |
> |--------------|-----------------|------------------|--------|----------|---------|
> | DiffBIR      | 380.0M          | 10.27            | 12117  | 0.5664   | 69.87   |
> | SeeSR        | 749.9M          | 7.18             | 32928  | 0.5036   | 68.67   |
> | PASD         | 625.0M          | 4.58             | 14562  | 0.4371   | 67.78   |
> | ResShift     | 118.6M          | 1.13             | 2745   | 0.3693   | 58.90   |
> | OSEDiff      | 8.5M            | 0.18             | 1133   | 0.4410   | 67.96   |
> | VARestorer (Ours) | 27.3M         | 0.23             | 1536   | **0.5590** | **72.32** |
>
> > ### **About Figure 3**
>
> **[Reply]:** Thank you for pointing this out. We have revised Figure 3 to eliminate the ambiguity: all labels are now explicitly defined (e.g., image encoder $\mathcal{E}$ and image decoder $\mathcal{D}$), and the arrows have been redrawn to clearly indicate the process flow and module interactions. We believe the updated figure resolves the confusion.
>
> > ### **About ablation on prompts**
>
> **[Reply]:** Thank you for the suggestion. We have conducted an ablation study where we removed the textual prompt during inference (empty text). The results (**Table A, w/o prompt**) show that using BLIP-generated prompts consistently **improves restoration quality**, confirming the effectiveness of text guidance in enhancing VARestorer’s performance:
> | Method             | LPIPS ↓   | MUSIQ ↑ | NIQE ↓  | CLIPIQA ↑ |
> |-------------------|-----------|---------|---------|-----------|
> | w/o prompt         | 0.3201    | 67.05   | 4.831   | 0.7332    |
> | VARestorer     | **0.3131**    | **72.32** | **4.410** | **0.7669** |
>
> > ### **About full tuning**
>
> **[Reply]:** Thanks for the suggestion. We have compared full fine-tuning with LoRA (Table A, full tuning):
> | Method             | LPIPS ↓   | MUSIQ ↑ | NIQE ↓  | CLIPIQA ↑ |
> |-------------------|-----------|---------|---------|-----------|
> | full tuning        | **0.3127**    | 70.58   | 4.976   | 0.6864    |
> | VARestorer    | 0.3131    | **72.32** | **4.410** | **0.7669** |
>
> Interestingly, full tuning slightly lowers metrics (e.g. CLIPIQA) and converges much more slowly (10K vs. > 30K), likely disturbing the pretrained generative prior. LoRA preserves most weights, adds lightweight task-specific adaptations, and achieves faster convergence with better performance. We have included a discussion about this finding in Section D.2.
>
> > ### **About teacher errors**
>
> **[Reply]:** Thank you for raising this point. In our design, the student is not forced to replicate potential teacher errors:
> - **Direct supervision with ground truth**: We include MES and perception loss between the student’s output and the GT image, ensuring the student is anchored to the true target rather than merely imitating the teacher.
> - **Teacher operates on HQ inputs**: The teacher receives the paired HQ image—not the degraded LQ input—so its predictions are far more reliable and less prone to artifacts.
> - **Errors do not accumulate**: Occasional discrepancies in the teacher's single-step forward pass act as mild training noise, which is far smaller and less harmful than the inference-time error accumulation in multi-step VAR.
>
> Together, these mechanisms prevent the student from inheriting systematic teacher errors while still benefiting from the teacher’s strong high-quality priors.
>
> >### **About limitations of one-step inference**
>
> **[Reply]:** We acknowledge this limitation and have added a discussion in **Section E** in the appendix, along with failure cases under extreme degradations. The one-shot formulation constrains the model’s generative capacity to a single step, which can hinder recovery of fine details when degradations are highly complex or unseen. In such cases, VARestorer cannot fully reconstruct missing or ambiguous signals.

---

### Official Review · Reviewer_BsAZ · 2025-10-29

**Soundness:** 3
**Presentation:** 4
**Contribution:** 3
**Rating:** 6
**Confidence:** 4

**Summary:**

This paper proposes a VAR-based distillation framework that distills the generative prior of a pretrained VAR model into a one-step model to achieve more efficient real-world image super-resolution. In addition, the authors propose a full-attention-based cross-scale conditioning mechanism to further improve generation quality. Experiments on multiple datasets demonstrate significant improvements on no-reference metrics.

**Strengths:**

1. The paper’s narrative is coherent, the language is clear and fluent, and it is easy to follow.

2. Beyond super-resolution, the authors also conduct experiments on de-raining and low-light enhancement, demonstrating the effectiveness of the method.

3. Through ablation studies, the paper validates the effectiveness of each component.

**Weaknesses:**

From Table 1, although the method shows clear gains on no-reference metrics, its performance on reference-based metrics is poor. This result may suggest a fidelity–perception trade-off; the authors should explain possible reasons for the weak reference-based performance.

**Questions:**

Why is a restorer used as a preprocessing step? How does the VAR model perform without it?

---

> ### Author Response · Authors · 2025-11-22
>
> We sincerely thank Reviewer BsAZ for the valuable insights and constructive feedback. Below, we address your concerns and questions in detail.
> >### **About reference-based performance**
>
> **[Reply]:** We agree that our method is relatively poor on traditional full-reference metrics (PSNR, SSIM, LPIPS). However, we believe this gap reflects a broader limitation of these metrics rather than a shortcoming of the proposed method. Metrics like PSNR and SSIM tend to **penalize high-frequency details**, and as restoration methods increasingly generate perceptually pleasing high-frequency content, pixel-wise or low-level similarity measures (PSNR/SSIM) and feature-distance proxies (LPIPS) **misalign with human perception— a point noted in prior work [a,b,c,d]**. In challenging scenarios, our outputs are visually more plausible and preferred by humans, whereas **methods with higher PSNR/SSIM often produce overly smooth or blurry images**. Qualitative comparisons in the manuscript (Figure 4) and the appendix (Figure B, Figure I) show that our outputs preserve fidelity to the input while producing richer, more vivid details than previous methods. This demonstrates that our model is **better suited for practical real-world image restoration tasks**. We provide further discussion on evaluation metrics in **Section D.6**.
>
> >### **About the restorer**
>
> **[Reply]:** We use the publicly available pretrained SwinIR restoration module [e] to coarsely remove simple degradations (e.g., noise), enabling more accurate textual prompt extraction. This two-stage design is common in recent image restoration pipelines and **helps the generative stage focus on high-frequency details** rather than low-frequency corrections [a, f, g], while also **improving the precision of prompt extraction**. Importantly, our method remains functional without SwinIR: removing the preprocessor causes only a **modest drop** in quantitative performance, and our method still outperforms prior work on most reported metrics. For transparency, we have added a quantitative ablation in **Table A** and **Section D.8** comparing results with and without SwinIR:
>
> | Method             | LPIPS ↓   | MUSIQ ↑ | NIQE ↓  | CLIPIQA ↑ |
> |--|----|-----|----|----|
> | w/o restorer     | 0.3150    | 72.27   | 4.458   | 0.7624    |
> | VARestorer | **0.3131**    | **72.32** | **4.410** | **0.7669** |
>
> In the future, we will also include brief implementation notes in the released code to ensure both settings can be easily reproduced.
>
> [a] Scaling up to excellence: Practicing model scaling for photo-realistic image restoration in the wild. CVPR, 2024.
>
> [b] The perception-distortion tradeoff. CVPR, 2018.
>
> [c] Pipal: a large-scale image quality assessment dataset for perceptual image restoration. ECCV, 2020.
>
> [d] Ntire 2022 challenge on perceptual image quality assessment. CVPR, 2022.
>
> [e] SwinIR: Image Restoration Using Swin Transformer. ICCV Workshop, 2021.
>
> [f] Diffbir: Towards blind image restoration with generative diffusion prior. ECCV, 2024.
>
> [g] InstaRevive: One-Step Image Enhancement via Dynamic Score Matching. ICLR, 2025.

---

> > ### Comment · Reviewer_BsAZ · 2025-11-28
> > **Response to the authors' feedback**
> >
> > My concerns have been well addressed and I am willing to keep my score. Thanks for the efforts.

---

### Official Review · Reviewer_YB72 · 2025-10-30

**Soundness:** 2
**Presentation:** 3
**Contribution:** 2
**Rating:** 4
**Confidence:** 4

**Summary:**

This paper proposes VARestorer, a one-step visual autoregressive (VAR) distillation framework for real-world image super-resolution (Real-ISR). The method addresses two key challenges in applying VAR to ISR: (1) the next-scale prediction mechanism's limited ability to exploit global low-quality context, and (2) error accumulation during iterative prediction. VARestorer distills a pre-trained text-to-image VAR model into a one-step model using distribution matching, and introduces cross-scale pyramid conditioning with full attention to better leverage input information. Experiments on DIV2K-Val, DrealSR, and RealSR show improvements on perceptual metrics (72.32 MUSIQ, 0.7669 CLIPIQA) with 10× faster inference than vanilla VAR.

**Strengths:**

1. This paper has clear problem motivation, Identifies specific limitations of VAR for ISR (error accumulation, limited context)
2. Solid experimental setup: LODO protocol, multiple datasets, standard metrics
3. Good visual quality: Figure 1, 4 show appealing results
4. Ablations validate components: Table 3 shows each component contributes
5. Practical efficiency: One-step inference is desirable for deployment

**Weaknesses:**

1. This paper has weak theoretical foundation, including the following points:
- KL divergence formulation (Eq. 3) with mismatched conditioning lacks justification
- No analysis of distillation convergence or optimality
- Cross-scale full attention's compatibility with autoregressive generation unclear
- Missing theoretical characterization of when method should succeed/fail

2. Insufficient experimental rigor:
- Training 1 epoch (10K steps) without convergence analysis is concerning
- Teacher fine-tuning on 512×512 (vs. original 1024×1024) impact not studied
- Baseline comparison potentially unfair (VARSR-10 vs VARestorer-1)
- No sensitivity analysis of loss weights (λKL, λperc, λMSE)
- "w/o distill" baseline terrible but not investigated

3. Method limitations underexplored:
- Caption dependency critical but only qualitatively discussed (Figure G), also failure on severe degradation acknowledged but not analyzed.
- Fixed resolution (512×512) limits practical utility
- PSNR/SSIM degradation vs. baselines suggests fidelity issues


4. For presentation:
- Parameter count inconsistencies (1.2% vs 0.09% vs 27.3M)
- Cross-scale conditioning implementation vague
- Missing algorithmic details for key components
- No discussion of why VAR is preferable to diffusion

**Questions:**

1. In Equation (3), pT and pS have different conditioning. How do you compute DKL(pT(rk | rHQ,<k) || pS(rˆk | rLQ))? Is this importance sampling, or are you assuming some form of equivalence?
2. How does "cross-scale full attention" preserve the autoregressive property of VAR? Don't future scales condition on past scales in VAR? Providing the attention mask explicitly would help.
3. Teacher fine-tuning: Why fine-tune the 1024×1024 VAR on 512×512? Doesn't this lose the high-resolution knowledge? Have you tried using the original teacher directly?
4. Training sufficiency: Why only 10K steps (1 epoch)? Have you verified convergence? What happens with more training?
5. Which is correct: 1.2% (abstract), 0.09% or 1.78M (Appendix A), or 27.3M (Table 2)?
6. Table 3 shows "w/o distill" performs very poorly. Does this mean VAR is inherently unsuitable for ISR without distillation?
7. Why does VARSR need 10 steps? Have you tried VARSR with 1 step for fair comparison?

---

> ### Author Response · Authors · 2025-11-22
> **Official Response to Reviewer YB72 (Part 1)**
>
> Thank you for your detailed review! We would like to extend our sincere gratitude for your rich and constructive suggestions, as well as your generous and valuable efforts to help us polish our work. Below, we provide responses to each concern and question. (As the reviewer raised numerous points, the responses are divided into several Official Comments due to space limitations.)
>
> > ### **About Equation (3)**
>
> **[Reply]:** Thank you for the question. Although $p_\mathcal{T}$ and $p_\mathcal{S}$ are conditioned on different inputs (HQ vs. LQ), this is intentional: the goal of the distillation is to transfer the teacher’s high‐quality generation behavior across scales to the student, which only sees the LQ input.
>
> In practice, we do not directly compute the KL divergence between two full conditional distributions under different conditions. Instead, for each training pair, we:
> - Feed the HQ image into the teacher to obtain its target tokens $r_k$ (with scale-wise causal mask).
> - Feed the corresponding LQ image into the student to obtain $\hat{r}_k$ (with cross-scale full mask).
> - Compute a token-level cross-entropy loss between $r_k$ and $\hat{r}_k$ as the result in Equation (3).
>
> This does not rely on importance sampling or any equivalence assumption between the two conditionings—the student simply learns to align its token predictions with the teacher’s outputs while being conditioned on low-quality inputs. We will provide a clearer description in the revision.
>
> > ### **About training sufficiency and convergence**
>
> **[Reply]:** Thank you for the question. VARestorer fine-tunes only 1.2% of the pre-trained VAR parameters, so it does not require long training to reach convergence. This is consistent with prior tuning-based ISR methods—for example, OSEDiff trains with batch size 16 for $\sim$20K steps (1 day on 4×A100). In comparison, our setting (8×L20, batch size 32) reaches convergence in 10K steps ($\sim$2 days, **3.7 epochs**). We include a loss-convergence curve in **Figure H** to verify this. Additional training beyond 10K steps yields negligible improvement, confirming that the model converges within our training schedule.
>
> > ### **About cross-scale full attention**
>
> **[Reply]:** Our cross-scale full attention intentionally differs from the original VAR’s scale-wise causal mask. In VAR, each scale attends only to past scales, which preserves an autoregressive ordering but **restricts cross-scale interaction**. For ISR, this masking leads to limited information flow and noticeably weaker restoration quality.
>
> In VARestorer, we remove this constraint and use a **full attention across all scales**, enabling richer feature fusion (the corresponding attention mask is shown in **Figure 3**)). Since our method performs single-step generation, preserving the autoregressive property is unnecessary. The full mask improves restoration quality while still retaining the pretrained VAR’s generative capacity.
>
> >### **About caption dependency and failure cases**
>
> **[Reply]:** Thank you for the comment. We agree that caption dependency is an important factor. To address this, we conducted an ablation (**Table A, w/o prompt**) where the textual prompt is removed during inference. Our results show that including the caption consistently improves perceptual quality and image fidelity, confirming its benefit：
> | Method    | LPIPS ↓   | MUSIQ ↑ | NIQE ↓  | CLIPIQA ↑ |
> |-----|----|---|----|-----|
> | w/o prompt   | 0.3201    | 67.05   | 4.831   | 0.7332    |
> | VARestorer  | **0.3131**    | **72.32** | **4.410** | **0.7669** |
>
> Regarding failures on severely degraded inputs, we have added more discussion in **Section E**. In these cases, the input contains insufficient information, and the model relies on its generative priors to produce plausible outputs. This can result in slight deviations from the original content but still produces realistic images that align with real-world distributions.
>
> These updates provide a more complete understanding of the role of captions and the limitations of our method under extreme degradation.
>
> > ### **About 1024px generation**
>
> **[Reply]:** Our model is not restricted to a fixed output resolution. While the main paper follows standard benchmarks and reports 512px results, we also discuss larger-scale restoration in **Section C** and provide qualitative examples in **Figure F**. To strengthen this part, we now include the loss curve for the 1024px experiment in **Figure H** and its quantitative performance in **Table A**, with additional discussion in **Section D.3**. The model produces strong results at 1024px, though it requires more tuning due to the increased resolution.
> | Method             | LPIPS ↓   | MUSIQ ↑ | NIQE ↓  | CLIPIQA ↑ |
> |--|----|-----|----|----|
> | 1024 resolution    | **0.3082**| 72.11   | 4.427   | 0.7645    |
> | VARestorer | 0.3131    | **72.32** | **4.410** | **0.7669** |
>
> **Responses continue in Part 2.**

---

> ### Author Response · Authors · 2025-11-22
> **Official Response to Reviewer YB72 (Part 2)**
>
> > ### **About VARSR-10**
>
> **[Reply]:** Thank you for the question. **VARSR strictly follows the original VAR sampling procedure**, which requires multi-step next-scale prediction. Unlike diffusion models, VAR does not allow changing the inference step schedule, so running VARSR with only one step is not feasible.
>
> Our method is fundamentally different: we introduce VAR distillation to compress the multi-step process into a single step. Even with this difference, **our one-step model still outperforms the 10-step VARSR**, showing the strength of our approach rather than an unfair comparison.
>
> > ### **About sensitive analysis of $\lambda$s**
>
> **[Reply]:** Thank you for the insightful suggestion. We conducted a sensitivity analysis on the loss weights, particularly varying $\lambda_{\text{KL}} \in \{0.05, 0.1, 0.3, 0.6\}$. As shown in **Figure H**, different values of $\lambda_{\text{KL}}$ mainly affect the convergence speed but do not change the final convergence point of the MSE loss.
> Furthermore, our broader hyperparameter search indicates that as long as the loss weights remain within reasonable ranges — $\lambda_{\text{KL}} \in [0.05, 0.8]$, $\lambda_{\text{MSE}} \in [0.3, 1]$, and $\lambda_{\text{perc}} \in [0.2, 1]$ — the final performance varies only marginally.
> Based on these observations, we choose $\lambda_{\text{KL}} = 0.1$, $\lambda_{\text{MSE}} = 0.5$, and $\lambda_{\text{perc}} = 0.25$ as a relatively optimal combination. We have added this discussion in **Section D.7**.
>
> > ### **About performance of "w/o distill"**
>
> **[Reply]:** Thank you for the comment. The poor performance of “w/o distill” does not mean VAR is unsuitable for ISR. This variant uses a simple ControlNet-based conditioning, which works well for diffusion models but is not well aligned with VAR’s discrete tokens and next-scale prediction. The **mismatch in conditioning design** is the main reason for the degradation. Prior work (e.g., VARSR) and our VARestorer show that VAR can perform well for ISR when paired with an appropriate conditioning and training strategy. We will update this discussion in the revision.
>
> >### **About PSNR/SSIM and fidelity**
>
> **[Reply]:** We agree that our method scores slightly lower on traditional full-reference metrics (PSNR, SSIM, LPIPS). However, this reflects a known limitation of these metrics rather than a flaw in our approach. As restoration methods increasingly generate high-frequency, perceptually rich details, pixel- or feature-based metrics often **misalign with human perception**. Qualitative results in **Figure 4** and **Figure B** (appendix) show that VARestorer preserves overall fidelity while producing sharper, more vivid details than prior methods. In challenging cases, our outputs are more visually plausible and preferred by humans, whereas **methods with higher PSNR/SSIM often remain blurry**. This demonstrates that our model is **better suited for practical real-world image restoration tasks**. We have added a further discussion about the metrics in **Section D.6**.
>
> > ### **Why VAR instead of diffusion**
>
> **[Reply]:** Thank you for the question. Our goal is not to beat diffusion models but to **explore an alternative generative paradigm for ISR**. We choose VAR for several reasons:
> - **Architectural alignment with SR**. VAR’s iterative next-scale prediction **naturally mirrors the SR process**, making it well-suited for reconstructing fine structures. As shown in Table 1, this leads to strong detail recovery.
> - **Compatibility with autoregressive token models**. VAR shares the same discrete-token and autoregressive formulation as LLMs, making it easier to integrate into future multimodal or unified autoregressive architectures.
>
> Thus, our work investigates VAR as a compelling and complementary alternative to diffusion for ISR.
>
> > ### **About the parameter inconsistency**
>
> **[Reply]:** Thank you for pointing this out. The inconsistency was caused by an oversight during revision. We have carefully checked and corrected the numbers in the updated version. The correct value is **27.3M** learnable parameters, which corresponds to **1.2%** of the transformer parameters.

---

### Official Review · Reviewer_RnKJ · 2025-10-31

**Soundness:** 3
**Presentation:** 3
**Contribution:** 3
**Rating:** 4
**Confidence:** 4

**Summary:**

This paper proposes a distilled VAR framework for real-world image super-resolution. The method distills a single-step student model from a multi-step VAR teacher to achieve faster inference while maintaining visual quality. A multi-step guidance strategy is introduced to reduce accumulated errors during training. Experiments on several benchmarks demonstrate competitive no-reference metrics and visually sharp results compared with existing SR methods.

**Strengths:**

The paper is clearly written and tackles an important problem in real-world image super-resolution. The idea of distilling a multi-step VAR model into a single-step version is practical and well-motivated for efficiency. Experiments are thorough, covering multiple benchmarks, and the proposed method achieves strong results on several perceptual quality metrics.

**Weaknesses:**

- **Inconsistency in the core motivation and training design.**

The paper claims that multi-step VAR models suffer from cumulative errors, yet the proposed approach distills knowledge from a multi-step VAR teacher. Since the teacher’s outputs are already affected by the same multi-step bias, the student may inherit those errors rather than eliminating them. This raises a conceptual inconsistency between the stated problem and the actual training setup, making it unclear whether the proposed method truly mitigates error accumulation or simply compresses it into a single step.

- **Mismatch between metrics and visual realism.**
Although the method achieves significantly higher no-reference quality scores on RealSR, the visual results (e.g., the flower example in Fig. 4) exhibit unrealistic, hallucinated textures. The generated details appear sharp but not natural, suggesting that the improvement in perceptual metrics may come from artificial high-frequency patterns rather than faithful detail recovery. This discrepancy questions whether the claimed perceptual gains reflect genuine visual quality improvements.

If these concerns are properly addressed and clarified, I would be willing to consider raising my score.

**Questions:**

please see the weaknesses

---

> ### Author Response · Authors · 2025-11-22
>
> We sincerely thank Reviewer RnKJ for the thoughtful feedback and the opportunity to further clarify our work. We address the concerns point-by-point below:
> > ### **About core motivation and our design**
>
> **[Reply]:** We thank the reviewer for the thoughtful question. We agree that providing a clearer conceptual link between our motivation and design is important. Our reasoning can be summarized as follows:
>
> (1) **Problem identification**. We find that directly applying VAR to ISR leads to noticeable artifacts. Our analysis shows that these issues stem from **error accumulation** across VAR’s multi-step upsampling process. Moreover, the multi-step design results in **high inference latency**, which limits its practicality for real-world restoration. These two factors motivate us to develop a mechanism that both mitigates error accumulation and significantly reduces inference time.
>
> (2) **Motivation for one-step inference**: A single-step pipeline naturally eliminates step-wise error accumulation, so we aim to explore how to adapt VAR into a one-step ISR framework.
>
> (3) **Why distribution matching with VAR teacher?** Rather than learning a direct LQ-HQ mapping via simple MSE loss (which is challenging due to the ill-posed nature of ISR), we want to simplify this learning by adopting soft distribution matching, which guides the student toward realistic image distributions. This can eases the learning burden and at the same time fully utilize the pre-trained VAR priors.
>
> (4) **Why teacher errors do not propagate?**. Although VAR as a teacher may produce occasional imperfections, these do not harm training because:
> - We combine **MSE and percepture loss** with the distribution matching loss, providing strong correction signals.
>
> - The teacher receives **GT tokens** at all scales, greatly reducing its prediction error.
>
> - Residual teacher errors occur only in a **single-step forward pass** during training, making them small and effectively behaving as benign training noise—far better than the multi-step error accumulation in vanilla VAR.
>
>
> Overall, with these designs, the student model avoids VAR’s inference-time error drift while still inheriting its rich multi-scale priors. Empirically, we do not observe degeneration or accumulated artifacts—our results consistently improve over multi-step VAR. This demonstrates that our framework effectively achieves its core motivation. We will expand this discussion in the revision.
>
>
> >### **About metrics and visual realism**
>
> **[Reply]:** Thank you for raising this concern! We agree that improvements in perceptual metrics should not come from artificial high-frequency patterns. In Figure 4, we show that our method produces substantially more detailed and realistic results compared to prior approaches. For example, in the flower case, most existing Real-ISR methods generate **only blurry or flattened petal boundaries**, while our model **recovers sharper edges and clearly distinguishable stamen structures**.
>
> We understand the reviewer’s concern that certain regions in the flower in Figrue 4 may appear slightly “AI-like” or overly high-frequency. We would like to clarify that such effects mainly arise from the inherent generative characteristics of the pre-trained VAR model. Importantly, **this behavior is often desirable in real-ISR settings**: when the input is heavily degraded and the original details are irretrievably lost, any method—diffusion, VAR, or GAN-based—must rely on learned priors. In these situations, our model produces plausible and natural-looking textures that align with real image distribution. While this may lead to slight deviations from the exact ground truth, **it avoids the overly smooth and blurry outputs** typical of fidelity-oriented methods and results in images that are **more meaningful for real-world use**.
>
>
> Crucially, our improvements do not arise from indiscriminate addition of high-frequency patterns.  Instead, the model learns to **balance high-frequency detail generation with faithful low-frequency structure.** As shown in our additional qualitative comparisons in **Figure I**, our model can faithfully reconstruct both high-frequency details (e.g., flowers) and low-frequency structures (e.g., desert, sea), demonstrating that it does not rely on synthetic textures but achieves coherent restoration across frequency bands.
>
>
> Overall, our perceptual improvements reflect genuine visual quality gains, not artificial sharpening. Our framework effectively regulates VAR’s generative priors so they enhance both detail and structure without introducing unstable artifacts. We have added this discussion and more examples in **Section D.6 and Figure I**.

---

> > ### Comment · Reviewer_RnKJ · 2025-11-26
> >
> > I thank the authors for their detailed feedback and clarifications.
> >
> > Regarding Point 1:
> > I appreciate the authors' explanation regarding the distillation process. The clarification that the teacher model utilizes GT tokens (Teacher Forcing) effectively addresses my concern about error accumulation during inference. I am satisfied with this response.
> >
> > Regarding Point 2:
> > I understand the authors' argument concerning the "perception-distortion trade-off." However, I would like to invite the authors to discuss this further, specifically in the context of T2I-based priors.
> >
> > In my view, the cause of lower fidelity scores (such as the higher LPIPS on DRealSR and RealSR) differs between method types. For non-pretrained methods, poor fidelity usually stems from blurriness. However, for methods leveraging strong generative priors (like T2I), a high LPIPS score combined with excellent no-reference metrics often suggests that the model might be generating sharp but semantically different details (hallucinations) rather than strictly restoring the original content.
> >
> > Could the authors provide further analysis or visual evidence to clarify whether the higher LPIPS in these specific datasets is primarily due to sharpness, or if there is a noticeable degree of semantic drift/hallucination compared to the Ground Truth?

---

> > > ### Author Response · Authors · 2025-11-27
> > >
> > > Thank you for the insightful follow-up. We agree that, for pretrained generative methods, higher LPIPS may arise from high-frequency reconstruction and semantic drift, and this distinction deserves careful examination.
> > >
> > > To further investigate this issue, we added a comparison with a non-pretrained method, BSRGAN [a], (new **Table C** and **Figure J**). We show Table C below:
> > >
> > > | Dataset    | Method        | PSNR ↑ | SSIM ↑ | LPIPS ↓ | MANIQA ↑ | MUSIQ ↑ | NIQE ↓ | CLIPIQA ↑ | FID ↓ |
> > > |----------------|------------------|-------------|------------|------------|-------------|------------|------------|--------------|-----------|
> > > | DIV2K-Val  | BSRGAN        | **24.58** | **0.6269** | 0.3351 | 0.5071 | 61.20 | 4.7518 | 0.5247 | 44.23 |
> > > | DIV2K-Val  | VARestorer (Ours)  | 21.08 | 0.5355 | **0.3131** | **0.5590** | **72.32** | **4.410** | **0.7669** | **31.11** |
> > > | DrealSR    | BSRGAN        | **28.75** | **0.8031** | **0.2883** | 0.4878 | 57.14 | 6.5192 | 0.4915 | 155.63 |
> > > | DrealSR    | VARestorer (Ours) | 24.31 | 0.6894 | 0.3584 | **0.5638** | **69.49** | **5.494** | **0.7810** | **149.7** |
> > > | RealSR     | BSRGAN        | **26.39** | **0.7654** | **0.2670** | 0.5399 | 63.21 | 5.6567 | 0.5001 | 141.28 |
> > > | RealSR     | VARestorer (Ours) | 22.78 | 0.6453 | 0.3249 | **0.5655** | **71.37** | **4.763** | **0.7423** | **117.2** |
> > >
> > >  As shown, BSRGAN achieves better fidelity scores (PSNR/SSIM/LPIPS) primarily because its outputs become **overly smooth and blurry** under heavy degradation. This reduces feature-space distance to the GT but yields poorer perceptual quality. In contrast, our method restores **natural high-frequency textures** and **sharper structures**, which improves no-reference quality but increases LPIPS when the ground-truth details are already lost.
> > >
> > > Regarding hallucination, our observations are:
> > > - **Under light degradation**, our model does not exhibit semantic drift and performs similarly to non-pretrained methods (**Figure J**).
> > > - **Under heavy degradation**, all generative methods necessarily rely on learned priors when the signal is irreversibly lost. We view this not as **arbitrary hallucination** but as an inherent and desirable **feature of generative models**: they produce plausible, natural textures that align with real-world distributions, even if they cannot perfectly match the missing GT details.
> > >
> > > As illustrated in **Figure J**, our higher LPIPS on real-world datasets is mainly due to **realistic high-frequency variations** (e.g., leaf textures, window patterns) and **sharper edges**, rather than incorrect semantic content. Although these details are not fully faithful to the unavailable GT, we regard them as a reasonable and principled solution in heavily degraded scenarios where the missing information is irrecoverable.
> > >
> > > We have added this clarification, along with the quantitative/qualitative comparisons, in **Section D.6**. We hope this addresses the reviewer’s concerns and we welcome any further discussion.
> > >
> > > [a] Designing a practical degradation model for deep blind image super-resolution, ICCV 2021.

---

> > > > ### Comment · Reviewer_RnKJ · 2025-11-28
> > > >
> > > > I thank the authors for their comprehensive response and for conducting the additional experiments with BSRGAN.
> > > >
> > > > Regarding Point 2, the new quantitative comparison in Table C and the qualitative analysis in Figure J effectively clarify the source of the higher LPIPS scores. I agree with the authors' assessment that in cases of heavy degradation, generating plausible high-frequency details is a desirable trait for generative restoration models, even if it results in lower fidelity scores compared to the over-smoothing seen in non-pretrained methods. The distinction you made between arbitrary semantic drift and realistic texture restoration is convincing.
> > > >
> > > > The authors have adequately addressed my concerns regarding both the distillation process and the interpretation of fidelity metrics in the context of T2I priors. The added analysis in Section D.6 strengthens the paper significantly.
> > > >
> > > > In light of these clarifications and the improved quality of the manuscript, I am raising my rating to 6. I am unable to update the score in the system at this moment. I will proceed with the update when possible or ensure the Area Chair is notified of my decision.

---

> > > > > ### Author Response · Authors · 2025-11-28
> > > > > **Appreciation for the Reviewer RnKJ’s Feedback and Updated Assessment**
> > > > >
> > > > > Thank you very much for the positive feedback and for taking the time to re-evaluate our work. We are glad that the additional analysis, BSRGAN comparison, and clarifications in Section D.6 successfully addressed your concerns. Your constructive comments greatly improved the quality and clarity of the manuscript.
> > > > >
> > > > > We also appreciate your willingness to update the score. We understand that the submission system is currently experiencing issues. If the system becomes available again, we will kindly remind you so the update can be completed.
> > > > >
> > > > > Thank you again for your thoughtful review and support.

---

### Author Response · Authors · 2025-11-22

We sincerely thank all reviewers for their thoughtful feedback and constructive suggestions. We are encouraged by the positive reception of our work—Reviewer RnKJ and Reviewer YB72 described our approach as “well-motivated and practical” with “solid experiments,” and Reviewer BsAZ recognized its strong generalization ability to extended tasks, “demonstrating the effectiveness.” We have carefully addressed all concerns and clarified potential sources of confusion, **incorporating the corresponding revisions into the updated manuscript** (primarily in the appendix, highlighted in blue).

---

### Meta-Review · Area_Chair_aif6 · 2026-01-02

**Summary:**

This paper proposes an effective VAR distillation framework that distills a pre-trained VAR model into an efficient one-step model for real-world image super-resolution. The major concerns of this paper include the unclear motivation, discrepancies between the metrics and the visualization results, limited theoretical analysis, insufficient experimental evaluations, and missing efficiency analysis.

**Reviewer Concerns:**

The authors provides corresponding analysis and results to solve the concerns of reviewers. The concerns of unclear motivation, discrepancies between the metrics and the visualization results, insufficient experimental evaluations, and missing efficiency analysis are solved well.

**Reviewer Scores:**

As mentioned above, the authors solve the concerns well. However, the theoretical analysis should be provided. In addition, the limitation analysis should be discussed.

---

### Decision · Program_Chairs · 2026-01-26

Accept (Poster)